# Self-assembling manifolds in single-cell RNA sequencing data

Alexander J Tarashansky[1], Yuan Xue[1], Pengyang Li[1], Stephen R Quake[1,2,3], Bo Wang[1,4]*

[1]Department of Bioengineering, Stanford University, Stanford, United States; [2]Department of Applied Physics, Stanford University, Stanford, United States; [3]Chan Zuckerberg Biohub, San Francisco, United States; [4]Department of Developmental Biology, Stanford University School of Medicine, Stanford, United States

**Abstract** Single-cell RNA sequencing has spurred the development of computational methods that enable researchers to classify cell types, delineate developmental trajectories, and measure molecular responses to external perturbations. Many of these technologies rely on their ability to detect genes whose cell-to-cell variations arise from the biological processes of interest rather than transcriptional or technical noise. However, for datasets in which the biologically relevant differences between cells are subtle, identifying these genes is challenging. We present the self-assembling manifold (SAM) algorithm, an iterative soft feature selection strategy to quantify gene relevance and improve dimensionality reduction. We demonstrate its advantages over other state-of-the-art methods with experimental validation in identifying novel stem cell populations of *Schistosoma mansoni*, a prevalent parasite that infects hundreds of millions of people. Extending our analysis to a total of 56 datasets, we show that SAM is generalizable and consistently outperforms other methods in a variety of biological and quantitative benchmarks.
DOI: https://doi.org/10.7554/eLife.48994.001

*For correspondence:
wangbo@stanford.edu

**Competing interests:** The authors declare that no competing interests exist.

## Introduction

Single-cell RNA sequencing (scRNAseq) datasets typically contain tens of thousands of genes, although many of them may not be informative for differentiating between cell types or states. Feature selection is thus commonly used to select a subset of genes prior to downstream analyses, such as manifold reconstruction and cell clustering (*Crow et al., 2018*; *Satija et al., 2015*; *Vallejos et al., 2015*). However, current approaches have two major limitations.

First, feature selection methods filter genes based on arbitrarily or empirically chosen thresholds, small changes in which may result in different gene sets (*Vallejos et al., 2017*). In addition, the selection of features typically operates under the assumption that genes with highly variable expression between individual cells capture biologically meaningful variation. Because single-cell transcriptomes are inevitably contaminated by a combination of random transcriptional and technical noise (*Grün et al., 2014*), the variation in biologically relevant genes may be hard to distinguish from the background noise, especially when the differences between cell populations are subtle. Resolving these differences, or 'signals', is essential to study a variety of biological problems, including identifying cell subtypes (*Olsson et al., 2016*; *Treutlein et al., 2014*; *Lönnberg et al., 2017*; *Fincher et al., 2018*; *Baron et al., 2016*; *Schwalie et al., 2018*) and quantifying the effects of molecular perturbations to otherwise homogeneous populations of cells (*Lane et al., 2017*). In such datasets, only a small fraction of the genes, and therefore only a small fraction of the total variation, may contain the signals relevant for distinguishing cell types or cell states. Choosing these features without a priori knowledge remains an unmet computational challenge.

**eLife digest** New technologies have enabled scientists to closely examine the activity of individual cells. One increasingly popular technique to do this is called single-cell RNA-sequencing and it relies on the fact that although all cells in an organism carry the same DNA, different cell types use different genes. This technique is powerful but can struggle to identify meaningful distinctions between cell types, especially when the differences are subtle.

In single-cell RNA-sequencing, the messenger RNA (mRNA) copied from each gene is collected and counted, and usually the more a gene is copied the more active it is. Differences in gene activity (also called gene expression) between two cells often imply that they are different types of cells. However, since only an infinitesimal amount of mRNAs can be collected from a single cell, the counting is often inaccurate. In addition, the transient changes in gene expression can make cells of the same type have different gene expressions. These factors make it challenging to determine what genes are informative for distinguishing between cell types.

To address this problem, Tarashansky et al. have developed a computational approach called Self-Assembling Manifold (or SAM for short) to identify differences in gene expression that can lead to a better classification of cell types. First, SAM groups the cells randomly and looks for genes with different expression patterns between the groups. By looking at differences between groups instead of differences between individual cells, SAM is 'averaging out' individual differences within groups. SAM then uses the resulting information to re-classify the cells and start the process over again, taking the new groups and finding differences between them. SAM repeats these steps until the classification stops changing and becomes stable. SAM does not require any existing knowledge about cell types or gene expression, meaning it is unbiased and widely applicable. To test the usefulness of the algorithm, Tarashansky et al. used SAM to identify new cell types in the medically important parasitic worm *Schistosoma mansoni*, which infects hundreds of millions of people worldwide every year.

SAM can tell cell types apart better than existing approaches, and it can find meaningful differences in systems with a lot of meaningless variability as demonstrated by evaluating SAM's performance on 55 other datasets. The potential applications of this approach are many, including the creation of detailed cell atlases recording the different types of cells throughout entire organisms.

DOI: https://doi.org/10.7554/eLife.48994.002

The second limitation is that existing methods have been almost exclusively benchmarked on well-annotated, gold standard datasets with clearly distinguishable cell types (*Wang et al., 2017*; *Kiselev et al., 2017*; *Duò et al., 2019*; *Bahlo et al., 2018*). These datasets are not informative for distinguishing the performance between methods, because the differences between cell types are relatively straightforward to detect. However, evaluating the performance of feature selection and/or dimensionality reduction methods on datasets with more subtle signals is difficult as their ground truth labels are typically ambiguous or nonexistent.

To overcome the shortcomings of current feature selection approaches, here, we introduce the Self-Assembling Manifold (SAM) method, an unsupervised, 'soft feature selection' algorithm that iteratively rescales gene expressions to refine a nearest neighbor graph of cells until the graph converges to a stable solution. At each iteration, SAM assigns more weight to genes that are spatially variable across the constructed graph, and this weighted gene expression is then used to improve the next nearest neighbor assignment. SAM presents two advantages: it rescales all genes according to their weights, solving the problem of thresholding, and it prioritizes genes that are variable across the intrinsic manifold of the data rather than selecting genes that are variable across individual cells.

In order to better distinguish the performance between methods, we define a network sensitivity measure to identify datasets with subtle signals. With limited annotations in most high-sensitivity datasets, we introduce unsupervised graph-based metrics to quantify the degree of structure within the reconstructed manifolds for comparison between methods. In addition, we perform benchmarking using known ground truth labels on simulated datasets spanning a wide range of sensitivities by

introducing increasing levels of noise to well-annotated datasets. These analyses reveal that SAM consistently improves feature selection and cell clustering.

To demonstrate the utility of SAM in practice, we provide an in-depth analysis of two datasets that are challenging to analyze using existing methods: stem cells in a human parasitic worm, *Schistosoma*, and activated macrophages (*Lane et al., 2017*). We show that SAM can capture novel biology undetectable by other methods and validate these results with experimental evidence.

## Results

### The SAM algorithm

The SAM algorithm begins with a random *k*-nearest neighbor (kNN) graph and averages the expression of each cell with its *k* nearest neighbors: $C = \frac{1}{k}NE$, where $N$ is the directed adjacency matrix and $E$ is the gene expression matrix (*Figure 1a*). For each gene $i$, SAM computes a spatial dispersion factor of the averaged expressions $C_i$, which measures variation across neighborhoods of cells rather than individual cells (Materials and methods). These dispersions are used to calculate the gene weights, which then rescale the expression matrix: $\hat{E} = E\sqrt{W_D}$, where $W_D$ is a diagonal matrix with gene weights along the diagonal. Using the rescaled expressions $\hat{E}$, we compute a pairwise cell distance matrix and update the assignment of each cell's *k*-nearest neighbors accordingly. This cycle continues until the gene weights converge.

To demonstrate the implementation and utility of SAM, below we analyze a challenging dataset comprised of a few hundred relatively homogeneous stem cells isolated from *Schistosoma mansoni* (*Figure 1—figure supplement 1*), a widespread human pathogen (*Hoffmann et al., 2014*). Using a protocol we have established previously (*Wang et al., 2018*), these cells were collected by sorting dividing cells from juvenile parasites harvested from their mouse hosts at 2.5 weeks post infection. At this stage, the parasites use an abundant stem cell population (~15–20% of the total number of cells) for rapid organogenesis and growth (*Wang et al., 2013*; *Wang et al., 2018*). Testing several existing methods (*Wang et al., 2017*; *Kiselev et al., 2017*; *Satija et al., 2015*), we found that they were not able to identify distinct cell populations in this dataset. In contrast, SAM finds a stable solution independent of initial conditions (*Figure 1b*). A graph structure with clearly separated cell populations self-assembles through the iterative process (*Figure 1c*). In parallel, the gene weights converge onto the final weight vector. Eventually, only a small fraction of genes (~1%) are strongly weighted and useful for separating cell clusters, reflecting the inherent difficulty of analyzing this dataset.

*Figure 1d* shows that SAM iteratively improves a series of graph characteristics, including the network-average clustering coefficient (NACC), modularity, and Euclidean norm of the spatial dispersions (Materials and methods). The NACC and modularity quantify the degree of structure within the graphs – graphs with high NACC and modularity have regions of high density separated by regions of low density. The dispersion quantifies the spatial organization of gene expression – the higher the spatial dispersion the less uniformly distributed the gene expressions are along the graph. The final graph metrics are independent of initial conditions, which can start from a random graph or the output of an existing manifold reconstruction algorithm (e.g. Seurat, *Satija et al., 2015*). Importantly, we verified that SAM does not artificially boost these metrics in data that lack inherent structure: when applying SAM to a randomly shuffled expression matrix, none of these metrics increased from the random initial conditions.

### SAM identifies novel subpopulations within schistosome stem cells

Visualizing the converged graph in two dimensions using Uniform Manifold Approximation and Projection (*Becht et al., 2019*), we find that cells can be separated into three main populations, with Louvain clustering (*Blondel et al., 2008*) further splitting one of these clusters into two subpopulations (*Figure 2a*). In contrast, other commonly used dimensionality reduction methods, such as principal component analysis (PCA), Seurat (*Satija et al., 2015*), and SIMLR (*Wang et al., 2017*), failed to distinguish these cell populations (see Materials and methods for the selection of algorithms for comparison). Clustering the Seurat graph using Louvain clustering still results in a low-modularity partition and poor correspondence to the SAM cluster assignments.

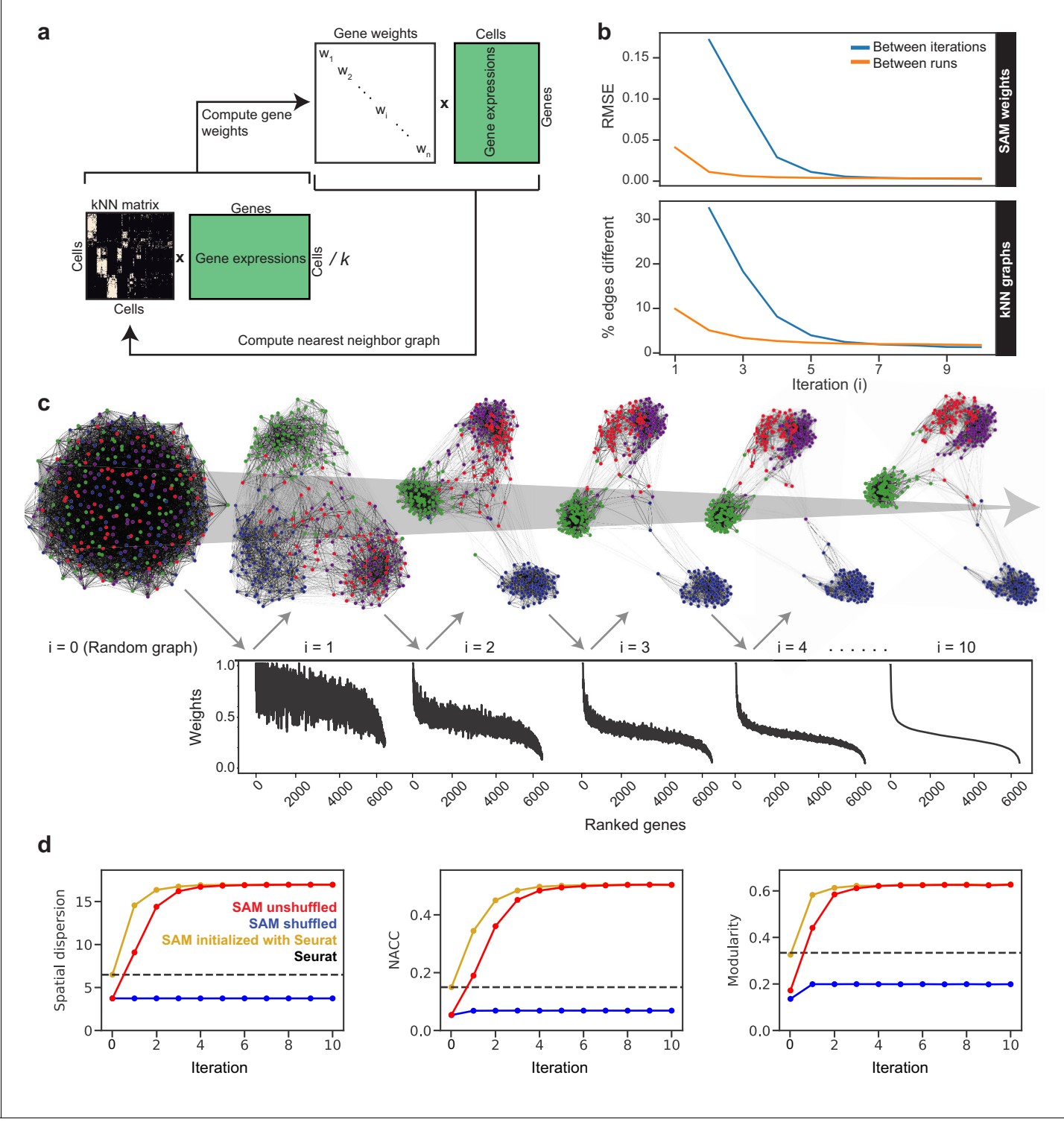

**Figure 1.** The SAM algorithm. (**a**) SAM starts with a randomly initialized kNN adjacency matrix and iterates to refine the adjacency matrix and gene weight vector until convergence. (**b**) Root mean square error (RMSE) of the gene weights (top) and the fraction of different edges of the nearest-neighbor adjacency matrices (bottom) between adjacent iterations (blue) and between independent runs at the same iteration (orange) to show that SAM converges to the same solution regardless of initial conditions. The differences between the gene weights and nearest-neighbor graphs from independent runs are relatively small, indicating that SAM converges to the same solution through similar paths. (**c**) Graph structures and gene weights of the schistosome stem cell data converging to the final output over the course of 10 iterations (*i* denotes iteration number). Top: nodes are cells and edges connect neighbors. Nodes are color-coded according to the final clusters. Bottom: weights are sorted according to the final gene rankings. (**d**)
*Figure 1 continued on next page*

*Figure 1 continued*

Network properties iteratively improve for the graphs reconstructed from the original data (red) but not on the randomly shuffled data (blue). The network properties converge to the same values when initializing SAM with the Seurat-reconstructed graph instead of a random graph (yellow). Dashed lines: metrics measured from the Seurat-reconstructed graphs.

DOI: https://doi.org/10.7554/eLife.48994.003

The following figure supplements are available for figure 1:

**Figure supplement 1.** Quality control of library preparation and sequencing of the schistosome stem cells.

DOI: https://doi.org/10.7554/eLife.48994.004

**Figure supplement 2.** A user interface for interactively exploring single-cell data using SAM.

DOI: https://doi.org/10.7554/eLife.48994.005

*Supplementary file 1* lists genes with high SAM weights, which includes most markers previously implicated to be enriched in subsets of schistosome stem cells (*Wang et al., 2013*; *Wang et al., 2018*). *Figure 2b* shows that the three populations include previously characterized δ'-cells, which specifically express an RNA binding protein *nanos-2* (Smp_051920), and ε-cells, which are marked by the expression of *eledh* (*eled*, Smp_041540) (*Wang et al., 2018*). More importantly, SAM reveals a novel stem cell population, μ, comprising ~30% of all sequenced cells (μ denotes muscle progenitors as discussed below). While μ-cells express ubiquitous stem cells markers (e.g. *ago2-1*, Smp_179320; *cyclin B*, Smp_082490) and cell cycle regulators (*Figure 2—figure supplement 1a*) (*Collins et al., 2013*; *Wang et al., 2013*; *Wang et al., 2018*), they are also strongly enriched for a large set of genes, with a calcium binding protein (*cabp*, Smp_005350), an actin protein (Smp_161920), an annexin homolog (Smp_074140), a helix-loop-helix transcription factor (*dhand*, Smp_062490), and a phosphatase (*dusp10*, Smp_034500) as the most specific markers of this population in comparison to other stem cells (*Figure 2—figure supplement 1b*).

Fluorescent in-situ hybridization (FISH) in conjunction with EdU labeling of dividing cells reveals that μ-cells (*cabp*$^+$EdU$^+$) are distributed near the parasite surface right beneath a layer of post-mitotic differentiated cells that also express *cabp* (*Figure 2c*). Close to the parasite surface, there are two major cell types intertwined in space: the skin-like tegumental cells and the body wall muscle cells. However, μ-cells express none of the recently identified markers in tegumental progenitors (*Wendt et al., 2018*), suggesting that they may be associated with the muscle lineage. To test this idea, we performed double FISH experiments and observed in post-mitotic *cabp*$^+$ cells the coexpression of a set of canonical muscle markers (*Witchley et al., 2013*), including tropomyosin (Smp_031770), myosin (Smp_045220), troponin (Smp_018250), and collagen (Smp_170340) (*Figure 2d*). These results suggest that *cabp* may mark the parasite body wall muscles and μ-cells are likely muscle progenitors, although functional validation is required to support this observation. Why the juvenile parasites maintain such an active pool of muscle progenitors will be an important question for future studies.

In addition, SAM identifies two subpopulations among ε-cells: ε$_α$-cells that are highly enriched for an aschaete-scute transcription factor (*astf*, Smp_142120), and ε$_β$-cells that abundantly express another basic helix-loop-helix protein (*bhlh*, Smp_087310) (*Figure 2b*, right panels). FISH experiments confirm these cells to be in close spatial proximity but with no coexpression of *astf* and *bhlh* (*Figure 2e*). Moreover, we observed with FISH that there are fewer *astf*$^+$ cells in larger, more matured juveniles, suggesting ε$_α$-cells are a dynamic population during development. To verify this observation, we sequenced another ~370 stem cell from juveniles at a later developmental time point (3.5 weeks post infection). After correcting for batch effects in the combined 2.5- and 3.5-week datasets using the mutual nearest neighbors (MNN) algorithm (*Haghverdi et al., 2018*), we find that δ'-, μ-, and ε$_β$-cells remain relatively constant throughout both time points, whereas ε$_α$-cells comprise a significantly smaller fraction of the stem cells at 3.5 weeks (7%) compared to 21% at 2.5 weeks (*Figure 2f*). Taken together, these analyses demonstrate that SAM can identify experimentally validated stem cell populations that are previously too subtle to separate using other methods but are closely associated with the schistosome development.

The critical difference between SAM and other methods lies in how they select genes for manifold reconstruction. SAM prioritizes genes with variable expressions across neighborhoods of cells rather than individual cells as in other methods (e.g. Seurat). *Figure 2g* shows that genes with high standardized dispersion across individual cells often have low SAM weights, indicating that these highly

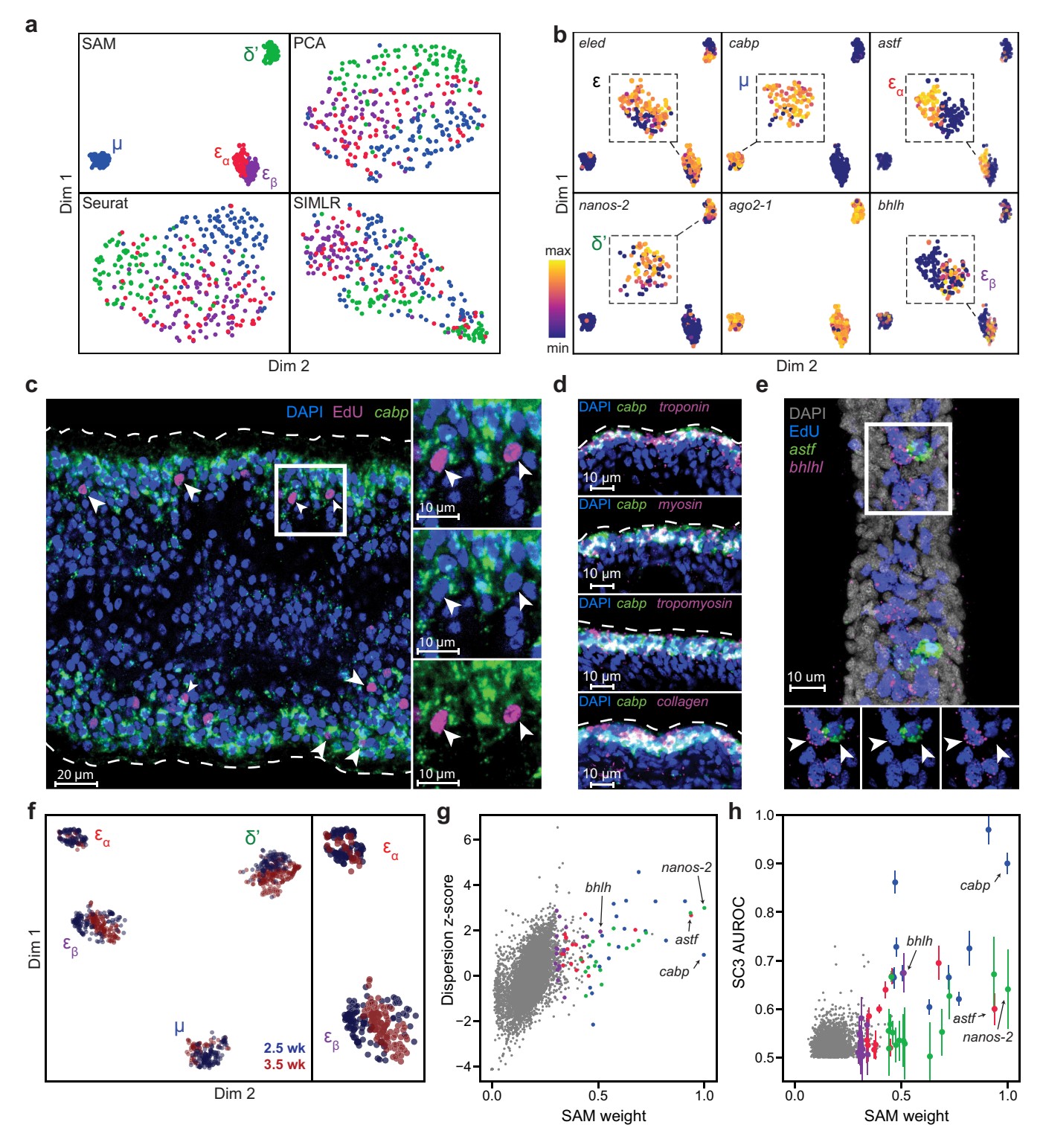

**Figure 2.** SAM identifies novel subpopulations within schistosome stem cells. (a) UMAP projections of the manifolds reconstructed by SAM, PCA, and Seurat. SIMLR outputs its own 2D projection based on its constructed similarity matrix using a modified version of t-SNE. The schistosome cells are color-coded by the stem cell subpopulations μ, δ', ε$_α$, and ε$_β$ determined by Louvain clustering. (b) UMAP projections with gene expressions of subpopulation-specific markers (*eledh, nanos-2, cabp, astf, bhlh,*) and a ubiquitous stem cell marker, *ago2-1,* overlaid. Insets: magnified views of the expressing populations. (c) FISH of *cabp* and EdU labeling of dividing stem cells in juvenile parasites at 2.5 weeks post-infection show that μ-cells

*Figure 2 continued on next page*

*Figure 2 continued*

(*cabp*⁺EdU⁺, arrowheads) are close to the parasite surface and beneath a layer of post-mitotic *cabp*⁺ cells. Dashed outline: parasite surface. Right: magnified views of the boxed region. (d) FISH of *cabp* and a set of canonical muscle markers, *troponin*, *myosin*, *tropomyosin*, and *collagen*, shows colocalization in post-mitotic *cabp*⁺ cells. Images in (c–d) are single confocal slices. (e) FISH of *astf* and *bhlh* shows their orthogonal expression in adjacent EdU⁺ cells (arrowheads). Bottom: magnified views of the boxed region. Image is a maximum intensity projection of a confocal stack with a thickness of 12 μm. (f) UMAP projection of stem cells isolated from juveniles at 2.5 and 3.5 weeks post-infection. Cell subpopulation assignments based on marker gene expressions are specified. Right: a magnified view to show the mapping of $\varepsilon_\alpha$- and $\varepsilon_\beta$-cells. (g) Standardized dispersions as calculated by Seurat plotted vs. the SAM gene weights. (h) SC3 AUROC scores plotted vs. the SAM gene weights. Error bars indicate the standard deviation of SC3 AUROC scores between trials using different chosen numbers of clusters. In (g) and (h), the top 20 genes specific to each subpopulation are colored according to the color scheme used in (a).

DOI: https://doi.org/10.7554/eLife.48994.006

The following figure supplement is available for figure 2:

**Figure supplement 1.** μ-cells express ubiquitous stem cell marker and population specific genes.

DOI: https://doi.org/10.7554/eLife.48994.007

variable genes (HVGs) are irrelevant to the topological relationships between cells. Other methods (e.g. SC3, *Kiselev et al., 2017*) identify marker genes based on differential gene expression between cell clusters, but this approach suffers when cell cluster assignment is poor, especially when discrete cell groups are difficult to separate or absent. Indeed, SC3 failed in the default mode as it incorrectly predicted there to be only one cluster in the schistosome dataset. After we manually increased the number of clusters, SC3 could recover a few of the marker genes associated with only one (μ-cells, blue symbols in *Figure 2h*) of the populations detected by SAM. Furthermore, changing the number of clusters resulted in different solutions and large variability in SC3 scores for its top ranked genes.

## SAM outperforms other state-of-the-art methods in extensive quantitative benchmarking

Below, we assess the general applicability of SAM by benchmarking its performance against state-of-the-art scRNAseq analysis methods on a large collection of datasets. We focus on three methods, that is, Seurat, SIMLR, and SC3, as they are mostly unsupervised, have been broadly used, and were shown to outperform other methods through extensive benchmarking (*Kiselev et al., 2017*; *Wang et al., 2017*; *Duò et al., 2019*; *Bahlo et al., 2018*; *Tian et al., 2019*). The criteria to select algorithms for comparison are explained in Materials and methods. We first benchmark against nine datasets (*Supplementary file 2*) that have high-confidence annotations to evaluate the accuracy of SAM in assigning cell clusters. Seven of these datasets are of pancreatic islet cells, as their subpopulations have been extensively characterized with known marker genes (*Baron et al., 2016*). For five out of the nine datasets, SAM has the highest Adjusted Rand Index (ARI, a measure of clustering accuracy) (*Hubert and Arabie, 1985*) with respect to the provided annotations (*Figure 3a*). On the remaining four Baron datasets, SAM and Seurat perform equally well with near perfect clustering accuracy, whereas SC3 and SIMLR tend to overestimate and underestimate the number of clusters, respectively. *Supplementary file 3* lists the clustering scores for each method and for each annotated cell type in the benchmarking datasets (Materials and methods). SC3 and SIMLR struggle to cleanly cluster cell types that constitute large fractions of the data, such as the alpha and beta cells in the pancreatic datasets. While Seurat performs well on the Baron datasets, it fails to identify alpha cells in the Wang and Muraro datasets when run with default parameters, although its performance is improved after optimizing parameters to maximize its clustering accuracy (Materials and methods). We note that this parameter optimization is impossible to perform on an experimental dataset with no available ground truth labels. Nevertheless, even with optimal parameters, Seurat has accuracy lower than or equal to that of SAM on all datasets.

SAM converges to the same set of gene weights for all datasets analyzed (*Figure 3b*, *Figure 3—figure supplement 1a*) and its performance is robust to the choice of parameters and random initial conditions (*Figure 3—figure supplement 1b–c*). In contrast, applying SAM to randomly generated datasets (Materials and methods), the resulting gene weights are highly dissimilar across random initial conditions (*Figure 3b*), indicating that SAM does not converge to a stable solution on datasets with no intrinsic structure. Finally, the scalability of SAM is similar to that of Seurat, capable of

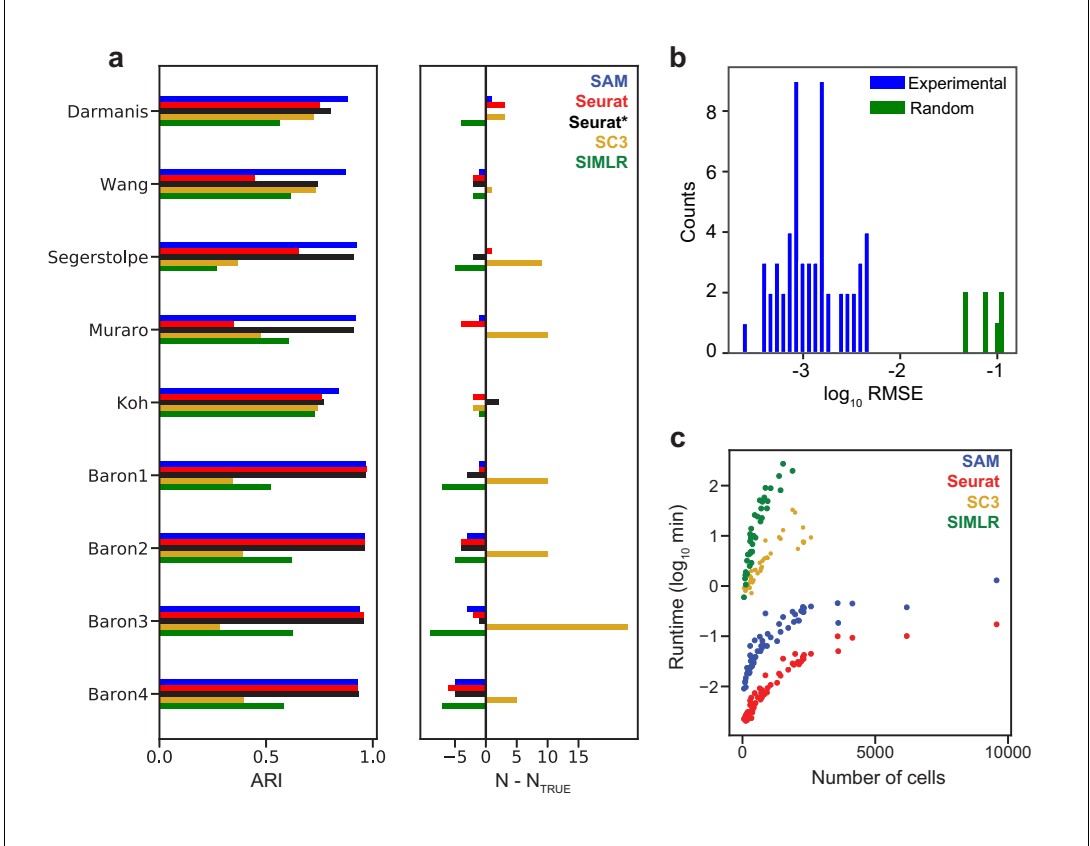

**Figure 3.** SAM improves clustering accuracy and runtime performance. (**a**) Accuracy of cluster assignment quantified by adjusted rand index (ARI) on nine annotated datasets (left). Right: differences between the number of clusters found by each method (N) and the number of annotated clusters ($N_{TRUE}$). Smaller differences indicate more accurate clustering. Seurat* denotes Seurat analysis using parameters that maximize ARI. (**b**) RMSE of gene weights output by SAM averaged across ten replicate runs with random initial conditions for 56 datasets (blue) and simulated datasets with no intrinsic structure (green, Materials and methods). (**c**) Runtime of SAM, SC3, SIMLR, and Seurat as a function of the number of cells in each dataset. SC3 and SIMLR were not run on datasets with >3000 cells as the run time exceeds 20 min.

DOI: https://doi.org/10.7554/eLife.48994.008

The following figure supplement is available for figure 3:

**Figure supplement 1.** SAM converges to a stable solution independent of random initial conditions and is robust to the number of nearest neighbors and choice of distance metric.

DOI: https://doi.org/10.7554/eLife.48994.009

analyzing hundreds of thousands of cells in minutes (*Figure 3c*), whereas SIMLR and SC3 are orders of magnitudes slower and thus excluded from further benchmarking which requires the analysis of many more datasets.

Because the nine benchmarking datasets are all comprised of clearly distinguishable cell types, they may not represent the performance of methods on other datasets that contain cell populations that are only subtly different. To identify such datasets, we introduce a network sensitivity metric that quantifies the changes in the cell-to-cell distances when randomly selecting a subset of features from the gene expression matrices (Materials and methods). High network sensitivity indicates that changes to the selected features strongly alters the resulting topological network. Networks that are robust to the selected features correspond to datasets that have many redundant signals or genes corroborating the network structure. In the datasets we compiled (*Supplementary file 2*), all broadly used benchmarking datasets have lower sensitivities whereas the schistosome dataset, which we have shown to be challenging to analyze, has the highest sensitivity (*Figure 4a*). The fraction of genes with large SAM weights (>0.5) is negatively correlated with the network sensitivity, suggesting that the biologically relevant variation in datasets with high sensitivity is captured by relatively fewer genes (*Figure 4b*). Analyzing all 56 datasets, we found that SAM improves the clustering,

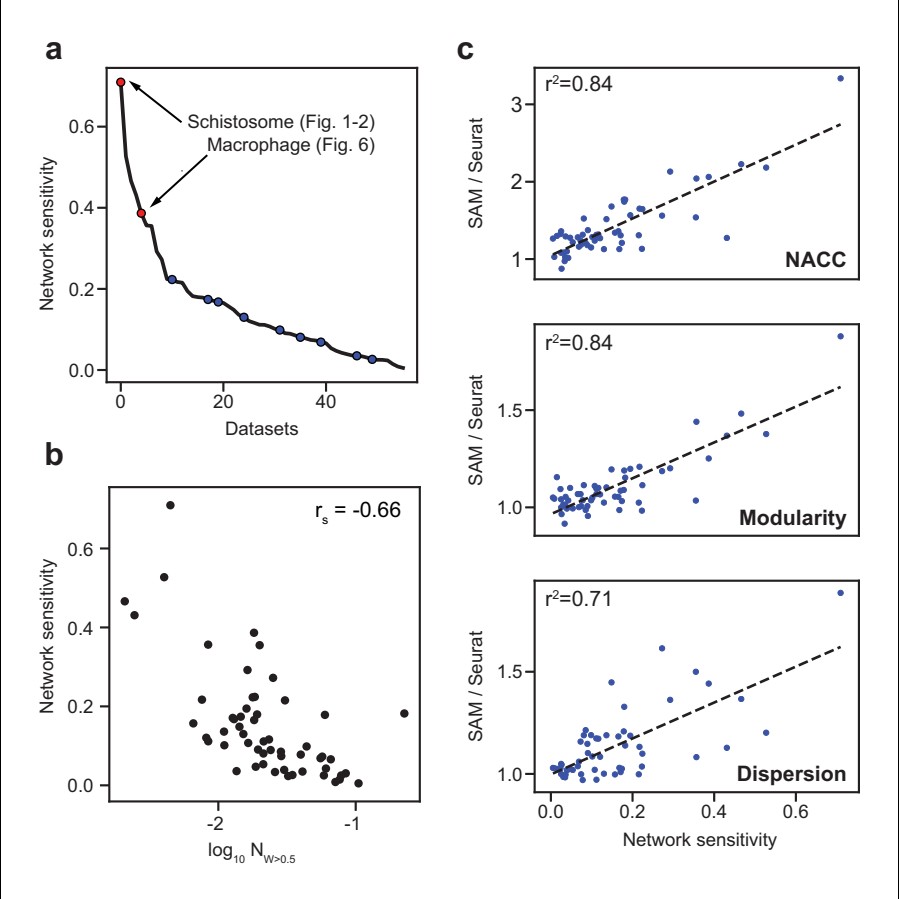

**Figure 4.** SAM improves the analysis of datasets with varying network sensitivities. (a) Network sensitivity of all 56 datasets ranked in descending order. Blue: the nine benchmarking datasets used in *Figure 3a*. Sensitivity measures the robustness of a dataset to changes in which features are selected (Materials and methods). (b) The network sensitivity plotted against the fraction of genes with SAM weight greater than 0.5 (in log scale) with Spearman correlation coefficient specified in the upper-right corner. (c) Fold improvement of SAM over Seurat for NACC, modularity, and spatial dispersion with respect to sensitivity for all 56 datasets. These ratios are linearly correlated with network sensitivity with Pearson correlations ($r^2$) specified in the upper-left corner of each plot.
DOI: https://doi.org/10.7554/eLife.48994.010

modularity, and spatial organization of gene expression across the graph in comparison to Seurat as the datasets become increasingly sensitive (*Figure 4c*).

Evaluating the clustering accuracy for the highly sensitive datasets, however, is challenging, because many of them have incomplete or nonexistent cell type annotations. Therefore, we use the nine well-annotated benchmarking datasets to simulate data across a wide spectrum of sensitivities. For this, we corrupt the data by randomly permuting gradually increasing fractions of the gene expressions. As illustrated by the Darmanis dataset (*Darmanis et al., 2015*), *Figure 5a* shows that the sensitivity increases along with the corruption. Below ~50% corruption, SAM's ARI scores only marginally decrease as the corruption (and thereby sensitivity) increases, whereas Seurat's performance rapidly deteriorates, even when run with optimal parameters. A similar contrast was observed between SAM and Seurat with the NACC, modularity, and spatial dispersion. Importantly, passing the genes with high SAM weights into Seurat rescued its performance across all metrics, indicating that SAM is able to consistently capture the genes relevant to the underlying structure of the data even with increasing levels of noise and illustrating the robustness of its feature selection strategy compared to the HVG filtering approach used by Seurat. These observations generalize to all nine benchmarking datasets, quantified by the area under the curves (AUC) of the metrics with respect to corruption (*Figure 5b*).

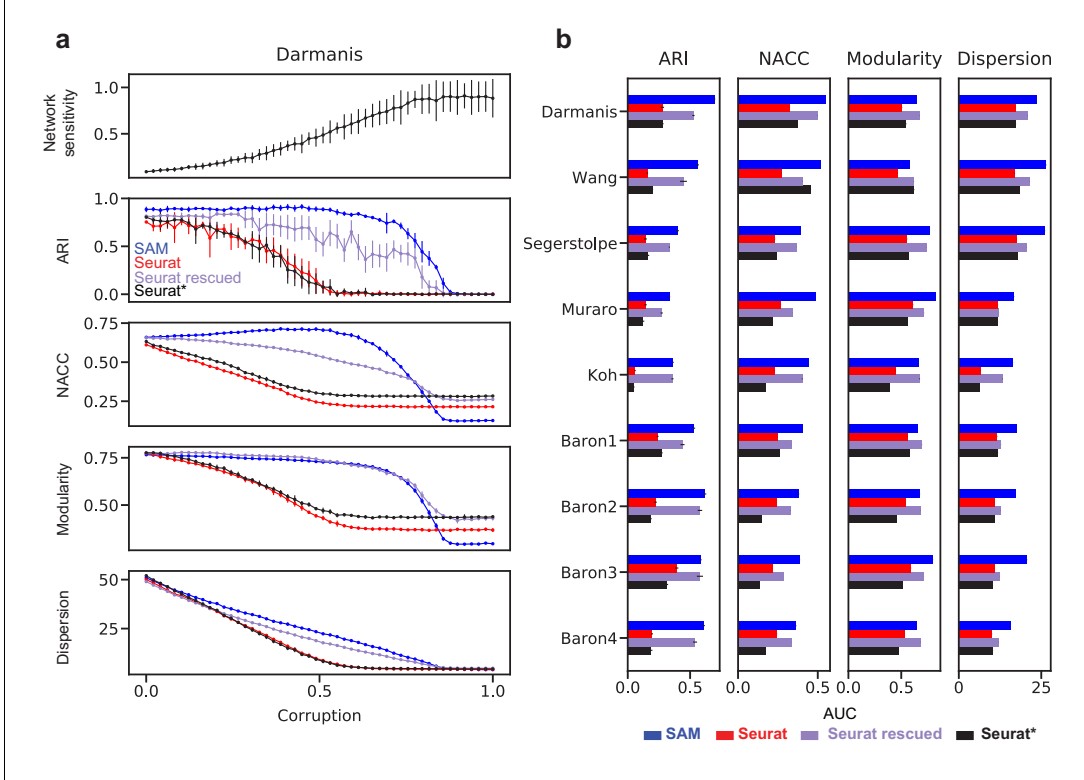

**Figure 5.** Robust feature selection improves cell clustering and manifold reconstruction. (a) Network sensitivity, ARI, NACC, modularity, and spatial dispersion with respect to corruption of the Darmanis dataset, in which we randomly permute fractions of the data ranging from 0 to 100% of the total number of elements (Materials and methods). Performance is compared between SAM (blue), Seurat (red), Seurat with optimal parameters (black), and Seurat rescued with the top-ranked SAM genes (indigo). Error bars indicate the standard deviations across 10 replicate runs. The errors for points with no bars are too small to be seen. (b) Comparison of the area under curve (AUC) of the metrics in (a) with respect to data corruption for all nine datasets. Error bars indicate the standard deviations across 10 replicate runs. The errors for data with no error bars are too small to be seen.
DOI: https://doi.org/10.7554/eLife.48994.011

## SAM clusters macrophages by their activation dynamics with proper temporal ordering

We next highlight another dataset to show that SAM can recover biologically meaningful information that other methods fail to capture. We chose this example, which contains ~600 macrophages treated with lipopolysaccharide (LPS) when individually trapped in microfluidic channels (*Lane et al., 2017*), because it has high network sensitivity (*Figure 4a*) and has accompanying single cell functional data of macrophage activation dynamics that may help to validate the results of our analysis. Applied to this dataset, SAM initially identifies two clusters (*Figure 6a*, top). Performing gene set enrichment analysis (GSEA, *Subramanian et al., 2005*), we find that genes with high SAM weights are dominated by cell cycle-related processes, with one of the clusters heavily enriched for cell cycle genes (e.g. Top2a, Mki67, *Figure 6—figure supplement 1a*). After removing the cell cycle effects (Materials and methods), SAM identifies two different clusters in which cells are properly ordered by the time since LPS induction, with the highly weighted genes being primarily involved in immune signaling (*Figure 6a*, bottom). These observations demonstrate that, in conjunction with GSEA, the quantitative gene weights output by SAM can be used to infer the biological pathways that drive the clustering of cells.

One of the two clusters is enriched for TNFα expression (*Figure 6b*). It is known that LPS activates two independent pathways, one through the innate immune signal transduction adaptor (Myd88) and the other through the TIR-domain-containing adapter-inducing interferon-β (TRIF) (*Lee et al., 2009*). While the Myd88 pathway directly activates NF-κB, the TRIF pathway first induces the secretion of TNFα, which subsequently binds to its receptor, TNFR, to prolong the activation of NF-κB

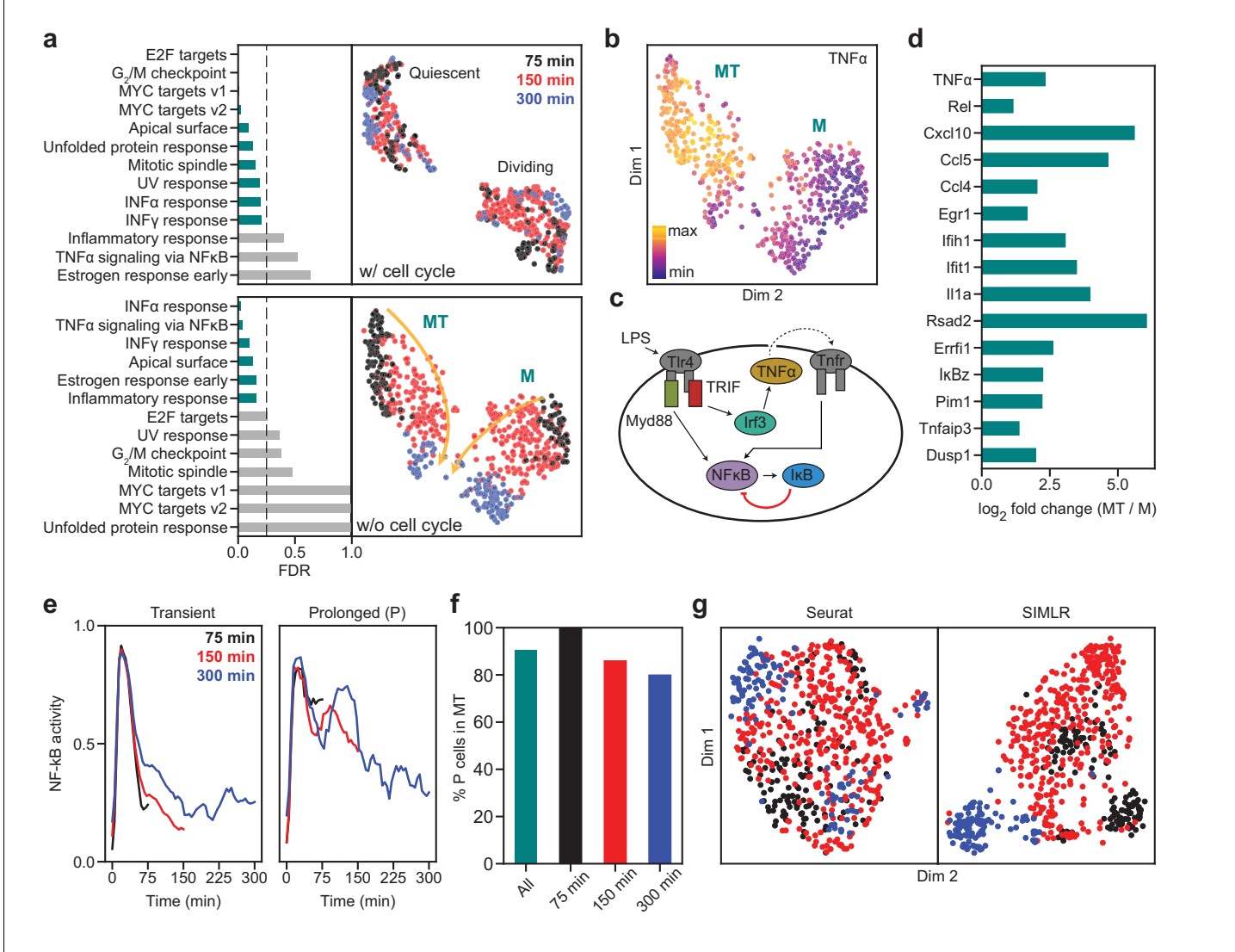

**Figure 6.** SAM captures the cellular activation dynamics in a stimulated macrophage dataset. (**a**) GSEA analysis (left) and UMAP projections (right) of the activated macrophages before (top) and after (bottom) removing cell cycle effects. Teal: significantly enriched gene sets determined by the significance threshold of 0.25 for the False Discovery Rate (FDR, dashed lines). Bottom: the two clusters are denoted as MT and M with colors representing the time since LPS induction. Arrows: evolution of time. (**b**) TNFα is enriched in the MT cluster. (**c**) Diagram of NF-κB activation in response to LPS stimulation via the Myd88 and TRIF signaling pathways. (**d**) Log$_2$ fold changes of the average expressions of selected inflammatory genes in the MT cluster vs. the M cluster. All genes are significantly differentially expressed between the two clusters according to the Welch's two-sample t-test ($p < 5 \cdot 10^{-3}$). (**e**) Representative traces for transient (left) and prolonged (right) NF-κB activation (Materials and methods). (**f**) Cells with prolonged NF-κB response (denoted as P) are primarily in the MT population. (**g**) Seurat and SIMLR projections show that they fail to order the cells by time since LPS induction and do not identify cell clusters representing the different modes of NF-κB activation.
DOI: https://doi.org/10.7554/eLife.48994.012

The following figure supplements are available for figure 6:

**Figure supplement 1.** Cluster-specific marker genes before and after removing cell cycle effects.
DOI: https://doi.org/10.7554/eLife.48994.013

**Figure supplement 2.** SAM groups cells based on NF-κB activation dynamics while other methods cannot.
DOI: https://doi.org/10.7554/eLife.48994.014

(*Figure 6c*). *Figure 6d* and *Figure 6—figure supplement 1b* show examples of genes that are highly enriched with TNFα, a number of which are inflammatory factors known to accumulate due to prolonged NF-κB activation (*Lane et al., 2017*). These results suggest that SAM grouped the cells based on their activated signaling pathways: one cluster is activated through both Myd88 and TRIF pathways (MT), while the other is only activated through Myd88 (M).

To further verify that the separation between the MT and M clusters truly reflects the dichotomy in cellular response to LPS induction, we noted that this dataset combines scRNAseq with live-cell imaging of NF-κB activity in single cells. This allows us to directly test if the MT and M clusters correspond to different signaling dynamics (Materials and methods). We found that most of the cells with prolonged NF-κB response (i.e. cells showing broad peaks of NF-κB activation in time) are in fact in the MT cluster (*Figure 6e–f*, and *Figure 6—figure supplement 2a*), consistent with the expectation that TNFα signaling prolongs NF-κB activation. Although our interpretation of the data matches that provided in the original study, we were able to analyze the dataset with almost no a priori knowledge. In contrast, the original study required extensive manual curation, analyzed only a subset of the dataset, and could not group cells by their NF-κB activation dynamics based on the gene expression data alone. Similarly, Seurat and SIMLR were unable to order the cells by the time since LPS induction or group cells based on their activation dynamics after removing the cell cycle effects (*Figure 6g*, and *Figure 6—figure supplement 2b–c*).

## Discussion

Here, we introduced a scRNAseq analysis method, SAM, which uses an unsupervised, robust, and iterative strategy for feature selection and manifold reconstruction. As demonstrated by our analysis of the schistosome stem cells and activated macrophages, SAM can capture biology that is undetectable by other methods. While SAM has consistently higher clustering accuracy than other state-of-the-art methods on datasets containing clearly distinct cell types, its advantages are especially apparent on datasets in which cell states or types are only distinguishable through subtle differences in gene expression.

The strength of SAM lies in the integration of three algorithmic components: spatial dispersion to measure feature relevance, soft feature selection, and the iterative scheme. By averaging the gene expression of a cell with that of its neighbors, the spatial dispersion quantifies the variation across neighborhoods of cells rather than individual cells. Genes with high spatial dispersion are more likely to be biologically relevant as they are capable of separating cells into distinct topological locations. Soft feature selection includes all genes and weights their contribution to the manifold reconstruction by their spatial dispersions. This mitigates the shortcoming of existing approaches in which the selection of features is a binary decision: genes are either included or not depending on arbitrarily chosen thresholds.

The conceptual challenge here is that calculating the gene weights requires the manifold, but reconstructing the manifold requires the gene weights for feature selection. SAM thus uses an iterative strategy to converge onto both the gene weights and the corresponding graph topology from a random initial graph. Each successive iteration refines the gene weights and network structure until the algorithm converges. Empirically, for all datasets analyzed we have shown that SAM converges onto a stable solution and is robust to the random initial conditions. Practically, it is possible to initialize SAM using the graph output of another method such as Seurat (*Figure 1d*), but using random initial conditions avoids potential biases in the analysis and enables the evaluation of the stability of SAM.

To demonstrate the strengths of SAM in practice, we analyzed the schistosome stem cells and identified novel stem cell populations that were validated by FISH experiments (*Figure 2*). In the analysis of activated macrophages, we showed that SAM can simultaneously order cells by the time since LPS induction and group cells according to their respective activated signaling pathways. We have validated this result using the live-cell imaging data presented in the original study (*Figure 6*).

We expect that the application of SAM is not limited to feature selection, cell clustering, and manifold reconstruction; it can be readily integrated with existing analytical pipelines as its gene weights and reconstructed manifolds can be used in downstream analyses. For example, we have shown how the genes ranked by their SAM weights can be used as input to GSEA to determine the biological processes enriched among the highly weighted genes (*Figure 6*), thus directly testing if

the weights reflect the biological relevance of genes. Additionally, the manifold reconstructed by SAM can be used as input to pseudotemporal ordering algorithms (*Setty et al., 2016*; *Trapnell et al., 2014*).

Beyond the two example case studies, we have rigorously evaluated SAM on a total of 56 datasets. While previous studies benchmarked on datasets with clearly defined cell populations, we defined a network sensitivity measure to rank the datasets based on the inherent difficulty of their analysis (*Figure 4*). Using these datasets, we showed that SAM consistently outperforms other methods in terms of both cell clustering accuracy measured by ground truth annotations, and manifold reconstruction measured by quantitative graph characteristics. These improvements can be attributed to the robust selection of features relevant for cell clustering and manifold reconstruction even in the presence of significant amounts of random noise, as shown in the corruption tests (*Figure 5*). Overall, the network sensitivity and quantitative benchmarking metrics should help in characterizing the performance of future scRNAseq analysis methods across a wider variety of datasets.

# Materials and methods

**Key resources table**

| Reagent type (species) or resource | Designation | Source or reference | Identifiers | Additional information |
|---|---|---|---|---|
| Commercial assay or kit | SsoAdvanced Universal SYBR Green Supermix | Biorad | 1725270 | qPCR |
| Commercial assay or kit | Quant-iT PicoGreen dsDNA Assay Kit | Thermo-Fisher | P7589 | cDNA quantification |
| Peptide, recombinant protein | RNase Inhibitor | Takara Bio | 2313B | RT mix |
| Chemical compound, drug | dNTP Set 100 mM solutions | Thermo-Fisher | R0181 | RT mix and cDNA pre-amplification |
| Sequence-based reagents | 100 µM oligo-dT | IDT | | AAGCAGTGGTATCAACGCAGAGTACT(30)VN |
| Sequence-based reagents | 100 µM TSO | Exiqon | | AAGCAGTGGTATCAACGCAGAGTACATrGrG+G |
| Commercial assay or kit | ERCC RNA Spike-In Mix | Thermo-Fisher | 4456740 | RT mix |
| Chemical compound, drug | 10% Triton X-100 | Thermo-Fisher | 28314 | RT mix |
| Peptide, recombinant protein | SMARTscribe reverse transcriptase | Takara Bio | 639538 | RT mix |
| Chemical compound, drug | 100 mM DTT | Promega | P1171 | RT mix |
| Chemical compound, drug | 5 M Betaine | Thermo-Fisher | B0300-1VL | RT mix |
| Commercial assay or kit | Kapa Hotstart Ready Mix | Roche | KK2602 | cDNA pre-amplification |
| Sequence-based reagents | 100 µM IS_PCR primer | IDT | | AAGCAGTGGTATCAACGCAGAGT |
| Peptide, recombinant protein | lambda exonuclease | NEB | M0262S | Depletion of primer dimers |
| Commercial assay or kit | Ampure purification beads | NEB | M0262S | DNA purification |
| Commercial assay or kit | TG Nextera XT DNA Sample Preparation Kit | Illumina | FC-131–1096 | Library preparation |
| Commercial assay or kit | TG Nextera XT Index Kit v2 Set A (96 Indices, 384 Samples) | Illumina | TG-131–2001 | Library preparation |

*Continued on next page*

*Continued*

| Reagent type (species) or resource | Designation | Source or reference | Identifiers | Additional information |
|---|---|---|---|---|
| Strain, strain background (*S. mansoni*) | NMRI | BEI Resources | NR-21963 | |
| Antibody | Anti-Digoxigenin-POD, Fab fragments from sheep | Roche | 11207733910 | (1:1,000); FISH experiments |
| Antibody | Anti-Fluorescein-POD, Fab fragments from sheep | Roche | 11426346910 | (1:1,500); FISH experiments |
| Peptide, recombinant DNA reagents | Plasmid-pJC53.2 | Addgene | 26536 | Cloning vector |
| Chemical compound, drug | Cy5-azide | Click Chemistry Tools | AZ118 | EdU detection |
| Chemical compound, drug | 5-ethynyl-2-deoxyuridine (EdU) | Invitrogen | A10044 | |
| Chemical compound, drug | Vybrant DyeCycle Violet (DCV) | Invitrogen | V35003 | FACS |
| Chemical compound, drug | TOTO-3 | Invitrogen | T3604 | FACS |

## Code and data availability

The SAM source code and tutorials can be found at https://github.com/atarashansky/self-assembling-manifold (*Tarashansky, 2019*; copy archived at https://github.com/elifesciences-publications/self-assembling-manifold). We have included a number of tutorials describing in detail the various functions, parameters, attributes, and data structures of the SAM package, and provided the documentation (docstrings) for all functions available to users. In addition, we have developed an interactive user interface that facilitates the convenient exploration of single-cell data and SAM parameters (*Figure 1—figure supplement 2*). A Jupyter notebook tutorial explaining how to use the interface is provided as well. The schistosome stem cell scRNAseq data generated in this study were obtained in two sequencing batches and are available through the Gene Expression Omnibus (GEO) under accession number GSE116920.

## Data processing

*Supplementary file 2* summarizes all datasets used in this study as well as the methods used to convert raw sequence read counts to gene expression, such as TPM (transcripts per million), CPM (counts per million), RPKM (reads per kilobase per million), or FPKM (fragments per kilobase per million). Datasets with asterisks next to their accession numbers are sourced from the *conquer* database (*Soneson and Robinson, 2018*). The nine benchmarking datasets used with high-confidence annotation labels are marked by crosses. Gene expression is measured in log space with a pseudocount of 1 (e.g. $\log_2(\text{TPM}+1)$). Genes expressed ($\log_2(\text{TPM}+1)>1$) in fewer than 1% or more than 99% of cells are excluded from downstream analysis as these genes lack statistical power. To reduce the influence of technical noise near the molecular detection limit, we set gene expression to zero when $\log_2(\text{TPM}+1)<1$. We denote the resulting expression matrix as $E$.

In the SAM algorithm (see below), we either standardize the gene expression matrix $E$ to have zero mean and unit variance per gene (which corrects for differences in distributions between genes) or normalize the expressions such that each cell has unit Euclidean (L2) norm (which prevents cells with large variances in gene expressions from dominating downstream analyses) prior to dimensionality reduction. In the below section, we denote the standardized or normalized expression matrix as $\bar{E}$. Empirically, we have found that standardization performs well with large, sparse datasets that are expected to contain many subpopulations, whereas L2-normalization is more suitable for smaller datasets with fewer subpopulations. This is likely due to the fact that standardization amplifies the relative expression of genes specific to small populations in large datasets, thereby making them easier to identify. In contrast, standardization decreases the relative expression of genes specific to

populations comprising larger fractions of the data, as is typically the case in smaller datasets, thereby making distinct populations more difficult to identify. *Supplementary file 2* documents the preprocessing step used for each dataset.

## The SAM algorithm

After first generating a random kNN adjacency matrix, the SAM algorithm goes through three steps that are repeated until convergence.

### Calculate the gene weights

First, the expression of each cell is averaged with its k-nearest neighbors:

$$C = \frac{1}{k}NE \tag{1}$$

where $N$ is the directed adjacency matrix for the kNN graph, and $E$ is the $n \times m$ log-transformed gene expression matrix with rows as cells and columns as genes. Here, we do not use $\bar{E}$ as it may contain negative values, for which dispersion is ill-defined. For each gene $i$, SAM computes the Fano factor from the averaged expressions $C_i$:

$$\mu_{C_i} = \frac{1}{n}\sum_{j=1}^{n} C_{ji} \tag{2}$$

$$\sigma^2_{C_i} = \frac{1}{n}\sum_{j=1}^{n}\left(C_{ji} - \mu_{C_i}\right)^2 \tag{3}$$

$$F_i = \frac{\sigma^2_{C_i}}{\mu_{C_i}} \tag{4}$$

where $\mu_{C_i}$ is the mean and $\sigma^2_{C_i}$ is the variance. We use the Fano factor to measure the gene expression variance relative to the mean in order to account for the fact that genes with high mean expressions tend to have higher variability. Computing the Fano factors based on the kNN-averaged expressions links gene dispersion to the cellular topological structure: genes that have highly variable expressions among individual cells but are homogeneously distributed across the topological representation should have small dispersions. $k$, set by default to 20, determines the topological length scale over which variations in gene expression are quantified. *Figure 3—figure supplement 1b* shows that the downstream analysis is robust to the specific choice of $k$. Additionally, the choice of $k$ does not significantly affect runtime complexity or scalability.

To compute the gene weights, we normalize the Fano factors to be between 0 and 1. First, we saturate the Fano factors to ensure that genes with large spatial dispersions do not skew the distribution of weights: $\{F_i | F_i > z\} = z$, where $z$ is the mean of the largest $N$ dispersions ($N = 50$ by default). We then calculate the gene weights as:

$$W_i = \frac{F_i}{z} \tag{5}$$

### Rescale the expression matrix

SAM multiplies the gene weights into the preprocessed expression matrix:

$$\hat{E} = \bar{E}\sqrt{W_D} \tag{6}$$

where $\bar{E}$ is the standardized or normalized expression matrix and $W_D$ is a diagonal matrix with $W_i$ along the diagonal. This matrix multiplication linearly rescales the gene expression variances and gene-gene covariances by their respective weights, attenuating the influence of genes with low dispersions across neighborhoods.

## Updating the kNN graph

To compute pairwise cell-cell distances, we perform PCA on the rescaled expression matrix $\hat{E}$. The variance-scaling operation in *Equation 6* improves the robustness of PCA to variations in genes that are uniformly distributed along the current graph (i.e. genes with low weights). Furthermore, this weighting strategy eliminates the typical requirement of selecting a subset of HVGs to feed into PCA, which often relies on arbitrary thresholds and heuristics. To perform PCA, we first mean center $\hat{E}$ to form $\hat{E}_\mu$:

$$\hat{E}_\mu = \hat{E} - \frac{1}{n}ee^T\hat{E} \tag{7}$$

where $e$ is a column vector of ones with dimension $n$. We then compute the Singular Value Decomposition (SVD) of $\hat{E}_\mu$:

$$\hat{E}_\mu = USV^T \tag{8}$$

with the principal components defined as

$$P = US \tag{9}$$

The eigenvalues corresponding to the eigendecomposition of the gene-gene covariance matrix are defined in terms of the singular values as

$$\Lambda = \frac{S^2}{n-1} \tag{10}$$

where $S$ is a diagonal matrix with singular values along the diagonal. Using the PC matrix $P$, SAM computes a pairwise cell-cell distance matrix. While typical dimension reduction approaches include a subset of the PCs, which is often subjective or requires computationally intensive maximum-likelihood approaches, we include all PCs and scale their variances by their corresponding eigenvalues:

$$\hat{P} = P\sqrt{\Lambda} \tag{11}$$

As a result, PCs with small eigenvalues are weighted less in the calculation of the distance between cells $i$ and $j$, $D_{\hat{P}_i\hat{P}_j}$. $D_{\hat{P}_i\hat{P}_j}$ is the Pearson correlation or Euclidean distance between rows $\hat{P}_i$ and $\hat{P}_j$ in the PC matrix. Pearson correlation distance is used by default, although *Figure 3—figure supplement 1c* shows that SAM is robust to the choice of distance metric. Using the distances to define the $k$-nearest neighbors for each cell, SAM updates the kNN matrix and repeats steps 1-3. The algorithm continues until convergence, defined as when the RMSE between gene weights in adjacent iterations converges:

$$\sqrt{\frac{1}{m}\sum_{j=1}^{m}\left(W_{i,j} - W_{i+1,j}\right)^2} < 5 \times 10^{-3} \tag{12}$$

where $m$ is the number of genes and $W_{i,j}$ is the weight for gene $j$ at iteration $i$.

## Visualization

To visualize the topological structure identified by SAM, we feed the final weighted PCA matrix, $\hat{P}$, into UMAP (*Becht et al., 2019*) using Pearson correlation as the distance metric by default. To directly visualize the final kNN adjacency matrix (*Figure 1c*), we used the Fruchterman-Reingold force-directed layout algorithm and drawing tools implemented in the Python package *graph-tool* (*Peixoto, 2017*).

## Choosing the benchmarking methods

We used three main criteria for choosing the benchmarking scRNAseq analysis methods: they should be widely used, have done extensive benchmarking against other methods, and be mostly unsupervised. We found on Web of Science that among the highest cited scRNAseq analysis tools in 2017–

2018 are Seurat, SC3, SIMLR, Reference Component Analysis (*Li et al., 2017*), Monocle (*Trapnell et al., 2014*; *Qiu et al., 2017*), zero-inflated factor analysis (ZIFA, *Pierson and Yau, 2015*), and Wishbone (*Setty et al., 2016*), of which we chose Seurat, SC3, and SIMLR.

SC3 is a consensus clustering algorithm that has done rigorous benchmarking against other methods such as SINCERA (*Guo et al., 2015*), SNN-Cliq (*Xu and Su, 2015*) and pcaReduce (*Žurauskienė and Yau, 2016*) on 12 datasets with ground truth annotation labels. SIMLR, a dimensionality reduction and clustering algorithm, evaluated its clustering performance on four annotated datasets against eight other dimensionality reduction methods, including PCA, Factor Analysis (FA), t-SNE, multidimensional scaling (MDS), and ZIFA. Both methods have demonstrated the highest clustering accuracy across most of the tested datasets. Additionally, as both methods have built-in functions to estimate the number of clusters present within the data, they are largely unsupervised. We also selected Seurat as one of the benchmarking methods, because it is arguably the most widely used tool for dimensionality reduction and clustering of scRNAseq data and has performed well in rigorous benchmarking studies against various methods including SC3, SIMLR, RCA, and pcaReduce (*Duò et al., 2019*; *Bahlo et al., 2018*).

We did not select Reference Component Analysis as it is primarily designed for cases in which an atlas of bulk, cell-type specific, reference transcriptomes is present. Additionally, we did not benchmark against Monocle and Wishbone, because they are pseudotime analysis methods and are meant for datasets with continuous branching processes such as cell differentiation. However, it is important to note that SAM can be used for dimensionality reduction upstream of pseudotime algorithms for such datasets. Finally, we did not benchmark against ZIFA as it has already been shown to have lower clustering accuracy than SIMLR.

In addition to measuring clustering accuracy, we also introduce the NACC, modularity, and spatial dispersion metrics to quantify both the degree of structure and spatial organization of gene expression within a nearest-neighbor graph. These metrics can only be applied to dimensionality reduction methods that construct a graph representation of the dataset. Consequently, we cannot use these metrics to evaluate SC3.

Although it does technically produce a graph representation of the data, SIMLR should be considered as a hybrid between a clustering and dimensionality reduction method. Because its similarity graph is assumed to have a block structure where the number of blocks is equal to the prespecified number of clusters, the resulting nearest-neighbor graph will, by construction, tend to have a higher degree of structure and therefore artificially inflated NACC and modularity.

Furthermore, the poor scalability of SC3 and SIMLR makes them difficult to run for many trials across a large number of datasets. Although SIMLR, in particular, does provide an alternative algorithm that can scale to run on much larger datasets, this alternative version has not been extensively benchmarked. Even so, despite the improved speed of this large-scale implementation, estimating the number of clusters using its built-in function remains a significant computational and memory bottleneck. For example, when applied to datasets with ~10,000 cells, neither implementations of SIMLR could estimate the number of clusters within 2 hr. As a result, we cannot run SIMLR in an unsupervised manner on datasets significantly larger than ~3000 cells.

As there are few practical alternatives for manifold reconstruction that have been extensively benchmarked and widely used, we primarily compare SAM to Seurat in tests involving the unsupervised, graph-based metrics to highlight the key, advantageous characteristics of SAM as a manifold reconstruction and feature selection algorithm when applied to datasets with varying sensitivities (*Figure 4a–c*).

## Benchmarking

To generate the convergence curves in *Figure 1b*, we computed the root mean square error (RMSE) of the gene weights averaged across all pairwise comparisons of ten replicates starting from randomly generated initial graphs. In *Figure 3b*, we extend this analysis to all datasets analyzed and report the final error. We use randomly generated datasets of varying sizes (ranging from 200 to 5000 cells) as a negative control to show that SAM does not converge onto the same solution across initial conditions when the data has no intrinsic structure. These datasets were randomly generated by sampling gene expressions from a Poisson distribution with mean drawn from a gamma distribution. To generate the convergence curves in *Figure 3—figure supplement 1a*, we computed the RMSEs, which are ensemble-averaged across ten replicate runs, between the gene weights in

adjacent iterations. We compute the adjacency error between kNN adjacency matrices $N_i$ and $N_j$ (*Figure 1b*) as

$$A_{i,j} = \frac{e^T |N_i - N_j| e}{2 e^T N_i e} \tag{13}$$

where $e$ is a column vector of ones. This simply measures the fraction of total edges that are different between the two graphs.

To compute the standardized dispersion factors in *Figure 2g*, we used Seurat's methodology implemented in Scanpy (*Wolf et al., 2018*), which groups the genes into 20 bins based on their mean expression values and computes the z-score of each gene's Fano factor with respect to the mean and standard deviation of all Fano factors in its corresponding bin. To generate the AUROC scores in *Figure 2h*, which quantify the likelihood of genes being cluster-specific markers, we ran SC3 on the schistosome data with the number of clusters ranging from 2 to 12. We used the AUROC scores corresponding to four clusters for the points on the scatter plot and the standard deviations of the scores across all tested numbers of clusters for the error bars.

We evaluated each analysis method on nine gold standard datasets (*Figure 3a*) using ARI, which measures the accuracy between two cluster assignments $X$ and $Y$ while accounting for randomness in the clustering:

$$ARI = \frac{\sum \binom{n_{ij}}{2} - \left[ \sum \binom{a_i}{2} \sum \binom{b_j}{2} \right] / \binom{n}{2}}{\frac{1}{2} \left[ \sum \binom{a_i}{2} + \sum \binom{b_j}{2} \right] - \left[ \sum \binom{a_i}{2} \sum \binom{b_j}{2} \right] / \binom{n}{2}} \tag{14}$$

where $n$ is the number of cells, and $n_{ij}$, $a_i$, and $b_j$ are elements from a contingency table that summarizes the overlap between the assignments $X$ and $Y$ (*Hubert and Arabie, 1985*). $n_{ij}$ denotes the number of cells assigned to $X_i$ that are also assigned to $Y_j$, while $a_i$ and $b_j$ are the sums of the $i$th row $j$th column of the contingency table, respectively. To calculate the clustering accuracy for each ground truth annotation label in *Supplementary file 3*, we decomposed the ARI into a vector of $j$ elements if $Y$ is the ground truth ($i$ otherwise) by not summing up the $j$ terms in the numerator, leaving it in vector form. Because the magnitudes of the cluster-specific scores depend on the number of cells in each cluster, a reference score was computed for each cluster using both $X$ and $Y$ as the true labels.

Seurat was implemented using the Scanpy package in Python (*Wolf et al., 2018*). For Seurat, we used both default and optimized parameters. In its default implementation, we selected the top 3000 variable genes according to their standardized dispersions and chose the number of PCs (bounded between 6 and 50) which explain 30% of the variance for dimensionality reduction. From these PCs, we calculated a cell-cell correlation distance matrix. To keep the comparison between SAM and Seurat graphs consistent, this distance matrix was converted into a kNN adjacency matrix with the value of $k$ used by SAM. We also ran a parameter sweep to optimize Seurat's performance for each benchmarking dataset separately by changing the number of highly variable genes and principal components to maximize the clustering accuracy.

To assign cluster labels for SAM and Seurat, we applied HDBSCAN (*McInnes et al., 2017*), an unsupervised, density-based clustering algorithm to their respective PCA outputs. As HDBSCAN does not cluster any cell it deems an outlier, we assign the remaining outlier cells to clusters using kNN classification. For each outlier cell, we identify its 20 nearest neighbors among the clustered cells. Outliers are assigned to the same cluster as that of the majority of its neighbors. This minor extension to HDBSCAN is available as the built-in function *hdbknn_clustering* in SAM. SC3 was run using default parameters. The SIMLR package was implemented in R and run with the normalization parameter set to 'True', which mean-centers gene expressions after normalizing them to be between 0 and 1. Both SC3 and SIMLR provide their own functions to estimate the number of clusters and cluster assignments.

To compare the quality of graphs generated by different methods, we use the NACC, modularity, and spatial dispersion. The NACC is the average of the local clustering coefficient for each node of a graph and quantifies the degree of structure in the graph (*Watts and Strogatz, 1998*). The local clustering coefficient is defined as

$$a_i = \frac{L_i}{k_i(k_i - 1)} \tag{15}$$

where $L_i$ is the number of edges between the $k_i$ neighbors of node $i$ and measures the degree of connectedness in a particular node's local neighborhood. We calculate the NACC using the implementation in *graph-tool* (**Peixoto, 2017**).

The modularity $Q$ of a graph is defined as

$$Q = \frac{1}{4m} \sum_{i,j}^{c} \left( A_{ij} - \frac{k_i k_j}{2m} \delta_{ij} \right) \tag{16}$$

where $A_{ij}$ is one if there is an edge from cell $i$ to cell $j$, $k_i$ is the degree of cell $i$, $k_j$ is the degree of cell $j$, $m$ is the total number of edges, and $\delta_{ij}$ is 1 if cells $i$ and $j$ are in the same cluster or 0 otherwise. High modularity indicates that clusters have on average more edges within clusters than between clusters. To find the optimal modularity for a particular graph, we used Louvain clustering, which searches for a partition with maximum modularity.

To quantify the spatial organization of gene expression along the graph, we calculate the Euclidean norm of the largest 100 spatial dispersions. Spatial dispersion is defined as before in the SAM algorithm: $F_i = \frac{\sigma^2_{C_i}}{\mu_{C_i}}$, where $F_i$ is the Fano factor of the kNN-averaged expressions and $C_i = \frac{1}{k} N E_i$. $N$ is the directed adjacency matrix output by SAM or Seurat and $E_i$ is a column vector of expression values for gene $i$.

To measure the inherent sensitivity of each dataset, we randomly perturbed the gene expression matrices of each dataset by randomly sampling 2000 genes and applied PCA to the subsampled data. A correlation distance matrix was calculated from the top 15 PCs and perturbations were repeated 20 times to generate distance matrix replicates. Sensitivity is then defined as the average error across all pairwise comparisons between replicates. The error between two distance matrices $j$ and $k$, $S_{jk}$, is defined as the average correlation distance between corresponding pairs of rows in the distance matrices $d_j$ and $d_k$:

$$S_{jk} = \frac{1}{n} \sum_{i=1}^{n} D\{d_{j,i}, d_{k,i}\} \tag{17}$$

where $D\{d_{j,i}, d_{k,i}\}$ is the Pearson correlation distance between the distances from cell $i$ in distance matrices $j$ and $k$.

We simulated datasets with increasing sensitivity by introducing increasing degrees of corruption in each of the nine annotated datasets. To corrupt a dataset, we randomly permuted a fraction $f$ of the elements in the expression matrix. The proportion of elements permuted corresponds to the degree of corruption, ranging from 0 to all elements. For each annotated dataset, we simulated 10 replicates per value of $f$. SAM and Seurat were run on each corrupted dataset, clustering was performed using the *hdbknn_clustering* function in SAM, and the ARI, NACC, modularity, and spatial dispersion metrics were recorded. The Area Under the Curve (AUC) was calculated for each metric with respect to $f$ using the trapezoidal rule. Finally, to rescue the performance of Seurat, we used as input to Seurat the top 3000 genes with the highest SAM weights.

## Gene set enrichment analysis (GSEA)

GSEA (**Subramanian et al., 2005**) is typically run on a gene expression matrix with user-defined cluster assignments to quantify the differential expression for each gene. By default, differential expression is quantified using a signal-to-noise metric and the resulting scores are used to rank the genes in descending order. However, GSEA can also run in an alternative mode in which the user provides a predefined list of gene rankings. Therefore, we used the genes ranked by their SAM weights as input to GSEA to determine the biological processes enriched among the highly weighted genes. As shown in *Figure 6a*, we can directly test if SAM captures the relevant biological processes. GSEA provides a number of statistical measures to assess the significance of enriched gene sets, of which we use the False Discovery Rate (FDR). The FDR quantifies the likelihood that a highly enriched gene

set represents a false positive. The significance threshold typically used with FDR is 25%, which implies that the results are likely to be valid 75% of the time.

## Removal of cell cycle effects

To remove cell cycle effects from the macrophage dataset, we adopted a simpler version of the strategy used in *ccRemover* (*Barron and Li, 2016*), in which we subtract from the data PCs that are significantly associated with known cell cycle genes. Letting $P$ represent the PCs and $L$ be the gene loadings, we quantify the association between the set of cell cycle genes $G$ and PC $j$ as

$$A_j = \frac{1}{|G|} \sum_{i \in G} |L_{ji}| \tag{18}$$

PC $j$ is selected if its association $A_j$ is at least two standard deviations above the mean of the associations for all PCs. In the particular case of the macrophage data, we identified the set of PCs $S = \{P_0, P_1, P_8\}$ as being significantly associated with the cell cycle genes. We next reconstruct the data using these PCs, which thus captures the cell-cycle effects, and subtract the reconstructed data from the expression matrix $E$:

$$E_{removed} = E - \sum_{j \in S} P_j L_j \sqrt{W} \tag{19}$$

When reconstructing the data, we scale the gene loadings by the SAM weights $W$ so that only the highly weighted SAM genes (which are initially dominated by cell cycle genes) contribute to the cell cycle removal, as there may be other genes involved in other biological processes that could also be correlated with the PCs in $S$. To run SAM on the data with cell cycle effects removed, we use $E$ as opposed to $E_{removed}$ for the calculation of spatial dispersions, because the latter may contain negative values, for which dispersion is ill-defined. This method is made available as a part of the SAM package in the functions *calculate_regression_PCs* and *regress_genes*.

## Clustering the NF-κB activity time series

The original study combined imaging and transcriptomics to link NF-κB nuclear translocation dynamics to changes in gene expression within single cells. Macrophages stimulated with LPS were individually trapped in microfluidic chambers and imaged for various lengths of time (75–300 min) prior to scRNAseq library preparation. NF-κB was tagged with a fluorescent protein, and its activation was measured as the nuclear-localized fluorescence intensity. Based on the imaging data, the authors identified three main classes of NF-κB dynamics, the first with a transient initial response, the second with a prolonged initial response, and the third with a recurrent response. Because the recurrent response is found only in the 300 min time point (the latest time point in the study) and comprises only ~8% of these cells, we primarily focused on clustering cells based on their initial dynamics. To do this, we used the *tslearn* (*Tavenard, 2017*) python package to group cells based on their NF-κB activity time series. Because these time series are quite noisy, we were conservative in labeling cells as having a prolonged initial response in an effort to avoid false positives. As a result, these cells comprise only ~30% of the dataset.

For the cells sampled at 75 and 150 min after LPS stimulation, we used the time series *k*-means algorithm with the *softdtw* distance metric to cluster them into three groups, which resulted in representative time series with transient, intermediate, and prolonged responses. Merging the cells with transient and intermediate responses into one cluster (which we simply labeled as transient response), we obtained the 75 and 150 min representative time series shown in *Figure 6e*. Because the cells sampled at 300 min displayed much more variability in their NF-κB time series, we clustered them into six groups, labeling the cluster whose representative time series had the broadest initial peak as the prolonged response cluster (blue in *Figure 6e*, right). The remaining groups were labeled as the transient response cluster (blue in *Figure 6e*, left).

## Correcting batch effects in the schistosome datasets

We used the Mutual Nearest Neighbors algorithm (*Haghverdi et al., 2018*) with default values to generate an expression matrix $E_{corrected}$ in which the batch effects between the 2.5-week and 3.5-

week datasets were corrected for. To run SAM on the batch-corrected data, we use $E$ for the calculation of spatial dispersions as opposed to $E_{corrected}$.

## scRNAseq of schistosome stem cells

Schistosome stem cells were isolated from juvenile parasites retrieved from infected mice at 2.5 and 3.5 weeks post infection. We followed the protocol as previously described (*Wang et al., 2018*). Briefly, we retrieved juvenile parasites from schistosome-infected mice (Swiss Webster NR-21963) by hepatic portal vein perfusion. Parasites were cultured at 37°C/5% $CO_2$ in Basch Medium 169 supplemented with 1X Antibiotic-Antimycotic for 24–48 hr to allow complete digestions of host blood cell in parasite intestines. In adherence to the Animal Welfare Act and the Public Health Service Policy on Humane Care and Use of Laboratory Animals, all experiments with and care of mice were performed in accordance with protocols approved by the Institutional Animal Care and Use Committees (IACUC) of Stanford University (protocol approval number 30366).

Before dissociation, parasites were permeabilized in PBS containing 0.1% Triton X-100% and 0.1% NP-40 for 30 s and washed thoroughly to remove the surfactants. The permeabilized parasites were dissociated in 0.25% trypsin for 20 min. Cell suspensions were passed through a 100 μm nylon mesh (Falcon Cell Strainer) and centrifuged at 150 g for 5 min. Cell pellets were gently resuspended, passed through a 30 μm nylon mesh, and stained with Vybrant DyeCycle Violet (DCV; 5 μM, Invitrogen), and TOTO-3 (0.2 μM, Invitrogen) for 30–45 min. As the stem cells comprise the only proliferative population in schistosomes, we flow-sorted cells at $G_2/M$ phase of the cell cycle on a SONY SH800 cell sorter. Dead cells were excluded based on TOTO-3 fluorescence. Single stem cells were gated using forward scattering (FSC), side scattering (SSC), and DCV to isolate cells with doubled DNA content compared to the rest of the population (*Wang et al., 2018*). Cells that passed these gates were sorted into 384-well lysis plates containing Triton X-100, ERCC standards, oligo-dT, dNTP, and RNase inhibitor.

cDNA was reverse transcribed and amplified on 384-well plate following the Smart-Seq2 protocol (*Picelli et al., 2013*). For quality control, we quantified the histone *h2a* (Smp_086860) levels using qPCR (the primers are listed in *Supplementary file 4*), as *h2a* is a ubiquitously expressed in all schistosomes stem cell (*Collins et al., 2013*; *Wang et al., 2013*; *Wang et al., 2018*). We picked wells that generated $C_T$ values within 2.5 $C_T$ around the most probable values (~45% of total wells, *Figure 1—figure supplement 1*). cDNA was then diluted to 0.4 ng/μL for library preparation. Tagmentation and barcoding of wells were prepared using Nextera XT DNA library preparation kit. Library fragments concentration and purity were quantified by Agilent bioanalyzer and qPCR. Sequencing was performed on a NextSeq 500 using V2 150 cycles high-output kit at ~1 million reads depth per cell. Raw sequencing reads were demultiplexed and converted to fastq files using bcl2fastq. Paired-end reads were mapped to *S. mansoni* genome version WBPS9 (WormBase Parasite) using STAR. In 2.5 week dataset, 338 cells with more than 1700 transcripts expressed at >2 TPM were used for downstream analysis. In the 3.5 weeks dataset, 338 cells with more than 1350 transcripts expressed at >2 TPM were used for downstream analysis (*Figure 1—figure supplement 1*).

## In situ hybridization and EdU labeling

RNA FISH experiments were performed as detailed in previous publications (*Collins et al., 2013*; *Wang et al., 2013*; *Wang et al., 2018*). Clones used for riboprobe synthesis were generated as described previously, with oligonucleotide primers listed in *Supplementary file 4*. Juvenile parasites were cultured with 10 μM EdU overnight, killed in 6 M $MgCl_2$ for 30 s, and then fixed in 4% formaldehyde with 0.2% Triton X-100% and 1% NP-40. Fixed parasites were sequentially dehydrated in methanol, treated in 3% $H_2O_2$ for 30 min, and rehydrated. Parasites were permeabilized by 10 μg/mL proteinase K for 15 min and post fixed with 4% formaldehyde. The hybridization was performed at 52°C with riboprobes labeled with either digoxigenin-12-UTP (Roche) or fluorescein-12-UTP (Roche). For detection, samples were blocked with 5% horse serum and 0.5% of Roche Western Blocking Reagent, and then incubated with anti-digoxigenin-peroxidase (1:1000; Roche) or anti-fluorescein peroxidase (1:1500; Roche) overnight at 4°C for tyramide signal amplification (TSA). For double FISH, the first peroxidase was quenched for 30 min in 0.1% sodium azide solution before the detection of the second gene. After FISH, EdU detection was performed by click reaction with 25 μM Cy5-azide conjugates (Click Chemistry Tools). Samples were mounted in *scale* solution (30%

glycerol, 0.1% Triton X-100, 4 M urea in PBS supplemented with 2 mg/mL sodium ascorbate) and imaged on a Zeiss LSM 800 confocal microscope.

## Acknowledgements

*S. mansoni* (strain: NMRI) was provided by the NIAID Schistosomiasis Resource Center for distribution through BEI Resources, NIH-NIAID Contract HHSN272201000005I. We thank F Zanini, N Neff, J Okamoto, and D Nanes-Sarfati for experimental help, K Lane, M Covert, J Qin, F Horns, G Stanley, J Rink, and M Chen for conceptual input and stimulating discussions. BW is supported by the Burroughs Wellcome Fund through the CASI program and a Beckman Young Investigator Award.

## Additional information

### Funding

| Funder | Grant reference number | Author |
|---|---|---|
| Burroughs Wellcome Fund | CASI program | Bo Wang |
| Arnold and Mabel Beckman Foundation | Young Investigator Award | Bo Wang |

The funders had no role in study design, data collection and interpretation, or the decision to submit the work for publication.

### Author contributions

Alexander J Tarashansky, Conceptualization, Data curation, Software, Formal analysis, Validation, Investigation, Visualization, Methodology, Writing—original draft, Writing—review and editing; Yuan Xue, Resources, Data curation, Formal analysis, Validation, Investigation, Visualization, Writing—review and editing; Pengyang Li, Validation, Investigation, Visualization; Stephen R Quake, Resources, Supervision, Funding acquisition, Writing—review and editing; Bo Wang, Conceptualization, Supervision, Funding acquisition, Methodology, Writing—original draft, Project administration, Writing—review and editing

### Author ORCIDs

Stephen R Quake (ID) https://orcid.org/0000-0002-1613-0809
Bo Wang (ID) https://orcid.org/0000-0001-8880-1432

### Ethics

Animal experimentation: In adherence to the Animal Welfare Act and the Public Health Service Policy on Humane Care and Use of Laboratory Animals, all experiments with and care of mice were performed in accordance with protocols approved by the Institutional Animal Care and Use Committees (IACUC) of Stanford University (protocol approval number 30366).

### Decision letter and Author response

Decision letter https://doi.org/10.7554/eLife.48994.125
Author response https://doi.org/10.7554/eLife.48994.126

## Additional files

### Supplementary files

• Supplementary file 1. Ranked gene list with high SAM weights in the schistosome stem cell data. Gene IDs and annotations are given in the *S. mansoni* genome version 9 (WormBase, WS268). Genes are assigned to the cluster corresponding to the marker gene, *nanos-2*, *cabp*, *astf*, or *bhlh*, with which they have the highest correlation. Genes found in our prior work (*Wang et al., 2018*) to be enriched in subsets of stem cells are specified.
DOI: https://doi.org/10.7554/eLife.48994.015

• Supplementary file 2. Datasets used in this study. Accession numbers, library size normalization methods, data preprocessing methods, sensitivity scores, and corresponding references are provided for each dataset. Accession numbers with asterisks indicate datasets that are sourced from the *conquer* database (**Soneson and Robinson, 2018**). Accession numbers with crosses indicate the nine well-annotated datasets that were used for benchmarking.
DOI: https://doi.org/10.7554/eLife.48994.016

• Supplementary file 3. ARI clustering accuracy of individual annotated cell types. The ARI scores of SAM, Seurat, SC3, and SIMLR applied to the nine benchmarking datasets are provided for each annotated ground truth cluster.
DOI: https://doi.org/10.7554/eLife.48994.017

• Supplementary file 4. Cloning primer sequences used for generating riboprobes for the FISH experiments and primer sequences for qPCR analysis. Functional annotations of the genes were given in the *S. mansoni* genome version 9 (WormBase, WS268).
DOI: https://doi.org/10.7554/eLife.48994.018

• Transparent reporting form
DOI: https://doi.org/10.7554/eLife.48994.019

## Data availability

The schistosome stem cell scRNAseq data generated in this study is available through the Gene Expression Omnibus (GEO) under accession number GSE116920.

The following dataset was generated:

| Author(s) | Year | Dataset title | Dataset URL | Database and Identifier |
|---|---|---|---|---|
| Xue Y, Wang B | 2018 | Single-cell RNA sequencing of proliferative stem cell population from juvenile Schistosoma mansoni | https://www.ncbi.nlm.nih.gov/geo/query/acc.cgi?acc=GSE116920 | NCBI Gene Expression Omnibus, GSE116920 |

The following previously published datasets were used:

| Author(s) | Year | Dataset title | Dataset URL | Database and Identifier |
|---|---|---|---|---|
| Tang F, Qiao J, Li R | 2013 | Single-cell RNA-Seq profiling of human preimplantation embryos and embryonic stem cells | https://www.ncbi.nlm.nih.gov/geo/query/acc.cgi?acc=GSE36552 | NCBI Gene Expression Omnibus, GSE36552 |
| Goolam M, Scialdone A, Graham SJL, Macaulay IC, Jedrusik A, Hupalowska A, Voet T, Marioni JC, Zernicka-Goetz M | 2016 | Heterogeneity in Oct4 and Sox2 targets biases cell fate in 4-cell mouse embryos | https://www.ebi.ac.uk/arrayexpress/experiments/E-MTAB-3321/ | ArrayExpress, E-MTAB-3321 |
| Tasic B, Menon V, Nguyen TN, Kim TK, Yao Z, Gray LT, Hawrylycz M, Koch C, Zeng H | 2016 | Adult mouse cortical cell taxonomy revealed by single cell transcriptomics | https://www.ncbi.nlm.nih.gov/geo/query/acc.cgi?acc=GSE71585 | NCBI Gene Expression Omnibus, GSE71585-GPL17021 |
| Guo F, Guo H, Li L, Tang F | 2015 | The transcriptome and DNA methylome landscapes of human primordial germ cells | https://www.ncbi.nlm.nih.gov/geo/query/acc.cgi?acc=GSE63818 | NCBI Gene Expression Omnibus, GSE63818 |
| Kim JK, Kolodziejczyk AA, Ilicic T, Illicic T, Teichmann SA, Marioni JC | 2015 | Single cell RNA-sequencing of pluripotent states unlocks modular transcriptional variation | https://www.ebi.ac.uk/arrayexpress/experiments/E-MTAB-2600/ | ArrayExpress, E-MTAB-2600 |
| Wollny D, Zhao S, Martin-Villalba A | 2016 | Single-cell analysis uncovers clonal acinar cell heterogeneity in the adult pancreas | https://www.ncbi.nlm.nih.gov/geo/query/acc.cgi?acc=GSE80032 | NCBI Gene Expression Omnibus, GSE80032 |
| Loh KM, Chen A, Koh PW, Deng TZ, Sinha R, TsaiJM, Barkal AA, Shen KY, | 2016 | Mapping the pairwise choices leading from pluripotency to human bone, heart, and other mesoderm cell types | https://www.ncbi.nlm.nih.gov/sra?term=SRP073808 | NCBI, SRP073808 |

| | | | | | |
|---|---|---|---|---|---|
| Jain R, Morganti RM, Shyh-Chang N, Fernhoff NB, George BM, Wernig G, Salomon REA, Chen Z, Vogel H, Epstein JA, Kundaje A, Talbot WS, Beachy PA, Ang LT, Weissman IL | | | | | |
| Deng Q, Ramsköld D, Reinius B, Sandberg R | 2014 | Single-cell RNA-seq reveals dynamic, random monoallelic gene expression in mammalian cells | https://www.ncbi.nlm.nih.gov/geo/query/acc.cgi?acc=GSE45719 | NCBI Gene Expression Omnibus, GSE45719 |
| Anoop P, Itay T | 2014 | Single-cell RNA-seq highlights intratumoral heterogeneity in primary glioblastoma | https://www.ncbi.nlm.nih.gov/geo/query/acc.cgi?acc=GSE57872 | NCBI Gene Expression Omnibus, GSE57872 |
| Rizvi AH, Camara PG, Kandror EK, Roberts TJ, Schieren I, Maniatis T, Rabadan R | 2017 | Single-cell topological RNA-seq analysis reveals insights into cellular differentiation and development | https://www.ncbi.nlm.nih.gov/geo/query/acc.cgi?acc=GSE94883 | NCBI Gene Expression Omnibus, GSE94883 |
| Tang Q, Langenau D | 2017 | Dissecting hematopoietic and renal cell heterogeneity in adult zebrafish at single-cell resolution using RNA sequencing | https://www.ncbi.nlm.nih.gov/geo/query/acc.cgi?acc=GSE100911 | NCBI Gene Expression Omnibus, GSE100911 |
| Engel I, Seumois G, Chavez L, Chawla A, White B, Mock D, Vijayanand P, Kronenberg M | 2016 | Innate-like functions of natural killer T cell subsets result from highly divergent gene programs | https://www.ncbi.nlm.nih.gov/geo/query/acc.cgi?acc=GSE74596 | NCBI Gene Expression Omnibus, GSE74596 |
| Edsgard D, Lanner F, Sandberg R, Petropoulos S | 2016 | Single-cell RNA-seq reveals lineage and X chromosome dynamics in human preimplantation embryos | https://www.ebi.ac.uk/arrayexpress/experiments/E-MTAB-3929/ | ArrayExpress, E-MTAB-3929 |
| Burns JC, Kelly MC, Hoa M, Morell RJ, Kelley MW | 2015 | Single-cell RNA-Seq resolves cellular complexity in sensory organs from the neonatal inner ear | https://www.ncbi.nlm.nih.gov/geo/query/acc.cgi?acc=GSE71982 | NCBI Gene Expression Omnibus, GSE71982 |
| Namani A, Wang XJ, Tang X | 2017 | Measuring signaling and RNA-Seq in the same cell links gene expression to dynamic patterns of NF-$\kappa$B activation | https://www.ncbi.nlm.nih.gov/geo/query/acc.cgi?acc=GSE94393 | NCBI Gene Expression Omnibus, GSE94383 |
| Biase FH, Cao X, Zhong S | 2014 | Cell fate inclination within 2-cell and 4-cell mouse embryos revealed by single-cell RNA sequencing | https://www.ncbi.nlm.nih.gov/geo/query/acc.cgi?acc=GSE57249 | NCBI Gene Expression Omnibus, GSE57249 |
| Trapnell C, Cacchiarelli D, Grimsby J, Pokharel P, Li S, Morse M, Mikkelsen T, Rinn J | 2014 | The dynamics and regulators of cell fate decisions are revealed by pseudotemporal ordering of single cells | https://www.ncbi.nlm.nih.gov/geo/query/acc.cgi?acc=GSE52529 | NCBI Gene Expression Omnibus, GSE52529-GPL16791 |
| Pollen AA, Nowakowski TJ, Shuga J, Wang X, Leyrat AA, Lui JH, Li N, Szpankowski L, Fowler B, Chen P, Ramalingam N, Sun G, Thu M, Norris M, Lebofsky R, Toppani D, Kemp DW, Wong M, Clerkson B, Jones BN, Wu S, Knutsson L, Alvarado B, Wang J, Weaver LS, May AP, Jones RC, Unger MA, Kriegstein AR, West JA | 2014 | Low-coverage single-cell mRNA sequencing reveals cellular heterogeneity and activated signaling pathways in developing cerebral cortex | https://www.ncbi.nlm.nih.gov/sra?term=SRP041736 | NCBI SRA, SRP041736 |
| Buettner F, Natar- | 2015 | Computational analysis of cell-to- | https://www.ebi.ac.uk/ar- | ArrayExpress, E- |

| | | | | |
|---|---|---|---|---|
| ajan KN, Casale FP, ProserpioV, Scialdone A, Theis FJ, Teichmann SA, Marioni JC, Stegle O | | cell heterogeneity in single-cell RNA-sequencing data reveals hidden subpopulations of cells | rayexpress/experiments/ E-MTAB-2805/ | MTAB-2805 |
| Satija R | 2014 | Single-cell RNA-seq reveals dynamic paracrine control of cellular variation | https://www.ncbi.nlm. nih.gov/geo/query/acc. cgi?acc=GSE48968 | NCBI Gene Expression Omnibus, GSE48968-GPL13112 |
| Ning L, Li-Fang C | 2015 | Oscope identifies oscillatory genes in unsynchronized single-cell RNAseq experiments | https://www.ncbi.nlm. nih.gov/geo/query/acc. cgi?acc=GSE64016 | NCBI Gene Expression Omnibus, GSE64016 |
| Meyer SE, Qin T, Muench DE, Masuda K, Venkatasubramanian M, Orr E, Paietta E, Tallman MS, Fernandez H, Melnick A, Beau MM, Kogan S, Salomonis N, Figueroa ME, Grimes HL | 2016 | DNMT3A haploinsufficiency transforms FLT3ITD myeloproliferative disease into a rapid, spontaneous, and fully penetrant acute myeloid leukemia | https://www.ncbi.nlm. nih.gov/geo/query/acc. cgi?acc=GSE77847 | NCBI Gene Expression Omnibus, GSE77847 |
| Treutlein B, Quake SR | 2014 | Reconstructing lineage hierarchies of the distal lung epithelium using single-cell RNA-seq | https://www.ncbi.nlm. nih.gov/geo/query/acc. cgi?acc=GSE52583 | NCBI Gene Expression Omnibus, GSE52583-GPL13112 |
| Olsson A, Venkatasubramanian M, Chaudhri VK, Aronow BJ | 2016 | Single-cell analysis of mixed-lineage states leading to a binary cell fate choice | https://www.ncbi.nlm. nih.gov/geo/query/acc. cgi?acc=GSE70245 | NCBI Gene Expression Omnibus, GSE70245 |
| Shin J, Song H | 2015 | Single-cell RNA-seq with waterfall reveals molecular cascades underlying adult neurogenesis | https://www.ncbi.nlm. nih.gov/geo/query/acc. cgi?acc=GSE71485 | NCBI Gene Expression Omnibus, GSE71485 |
| Schwalie PC, Dong H, Zachara M, Russeil J, Alpern D, Akchiche N, Caprara C, Sun W, Schlaudraff KU, Soldati G, Wolfrum C, Deplancke B | 2018 | A stromal cell population that inhibits adipogenesis in mammalian fat depots | https://www.ebi.ac.uk/ar-rayexpress/experiments/ E-MTAB-6677/ | ArrayExpress, E-MTAB-6677 |
| Darmanis S, Quake S | 2017 | Single-Cell RNA-Seq analysis of infiltrating neoplastic cells at the migrating front of human glioblastoma | https://www.ncbi.nlm. nih.gov/geo/query/acc. cgi?acc=GSE84465 | NCBI Gene Expression Omnibus, GSE84465 |
| Scialdone A, Tanaka Y, Jawaid W, Moignard V, Wilson NK, Macaulay IC, Marioni JC, Göttgens B | 2016 | Resolving early mesoderm diversification through single-cell expression profiling | https://www.ebi.ac.uk/ar-rayexpress/experiments/ E-MTAB-4079 | ArrayExpress, E-MTAB-4079 |
| Enge M, Arda HE | 2017 | Single-cell analysis of human pancreas reveals transcriptional signatures of aging and somatic mutation patterns | https://www.ncbi.nlm. nih.gov/geo/query/acc. cgi?acc=GSE81547 | NCBI Gene Expression Omnibus, GSE81547 |
| Stévant I, Neirjinck Y, Borel C, Escoffier J, Smith LB, Antonarakis SE, Dermitzakis ET, Nef S | 2018 | Deciphering cell lineage specification during male sex determination with single-cell RNA sequencing | https://www.ncbi.nlm. nih.gov/geo/query/acc. cgi?acc=GSE97519 | NCBI Gene Expression Omnibus, GSE97519 |
| Phillips MJ, Jiang P, Howden S | 2018 | A novel approach to single cell RNA-sequence analysis facilitates in silico gene reporting of human pluripotent stem cell-derived retinal cell types | https://www.ncbi.nlm. nih.gov/geo/query/acc. cgi?acc=GSE98556 | NCBI Gene Expression Omnibus, GSE98556 |

| Vanlandewijck M, He L, Mäe MA, Andrae J, Betsholtz C | 2018 | A molecular atlas of cell types and zonation in the brain vasculature | https://www.ncbi.nlm.nih.gov/geo/query/acc.cgi?acc=GSE99235 | NCBI Gene Expression Omnibus, GSE99235 |
|---|---|---|---|---|
| Furlan A, Dyachuk V, Kastriti ME, Calvo-Enrique L, Abdo H, Hadjab-Lallemend S, Chontorotzea T, Akkuratova N, Uso-skin D, Kamenev D, Petersen J, Suna-dome K, Memic F, Marklund U, Fried K, Topilko P, Lal-lemend F, Kharch-enko P, Ernfors P, Adameyko I | 2017 | Multipotent peripheral glial cells generate neuroendocrine cells of the adrenal medulla | https://www.ncbi.nlm.nih.gov/geo/query/acc.cgi?acc=GSE99933 | NCBI Gene Expression Omnibus, GSE99933 |
| Ghahramani A, Watt F, Luscombe N | 2018 | Epidermal Wnt signalling regulates transcriptome heterogeneity and proliferative fate in neighbouring cells | https://www.ncbi.nlm.nih.gov/geo/query/acc.cgi?acc=GSE99989 | NCBI Gene Expression Omnibus, GSE99989 |
| Lescroart F, Wang X, Li X, Gargouri S, Moignard V, Du-bois C, Paulissen C, Göttgens B, Blan-pain C | 2018 | Defining the earliest step of cardiovascular lineage segregation by single-cell RNA-seq | https://www.ncbi.nlm.nih.gov/geo/query/acc.cgi?acc=GSE100471 | NCBI Gene Expression Omnibus, GSE100471 |
| Mohammed H, Hernando-Herraez I, Reik W | 2017 | Single-cell landscape of transcriptional heterogeneity and cell fate decisions during mouse early gastrulation | https://www.ncbi.nlm.nih.gov/geo/query/acc.cgi?acc=GSE100597 | NCBI Gene Expression Omnibus, GSE100597 |
| Mathys H, Gao F, Tsai L | 2017 | Temporal tracking of microglia activation in neurodegeneration at single-cell resolution | https://www.ncbi.nlm.nih.gov/geo/query/acc.cgi?acc=GSE103334 | NCBI Gene Expression Omnibus, GSE103334 |
| Chevée M, Ro-bertson JD, Can-non GH, Brown SP, Goff LA | 2018 | Variation in activity state, axonal projection, and position define the transcriptional identity of individual neocortical projection neurons | https://www.ncbi.nlm.nih.gov/geo/query/acc.cgi?acc=GSE107632 | NCBI Gene Expression Omnibus, GSE107632 |
| Hook PW, MyCly-mont SA, Cannon GH, Law WD, Morton AJ, Goff LA, McCallion AS | 2018 | Single-Cell RNA-Seq of mouse dopaminergic neurons informs candidate gene selection for sporadic parkinson disease | https://www.ncbi.nlm.nih.gov/geo/query/acc.cgi?acc=GSE108020 | NCBI Gene Expression Omnibus, GSE108020 |
| Mi D, Li Z, Li M | 2018 | Early emergence of cortical interneuron diversity in the mouse embryo | https://www.ncbi.nlm.nih.gov/geo/query/acc.cgi?acc=GSE109796 | NCBI Gene Expression Omnibus, GSE109796 |
| Zanini F, Pu S, Bekerman E, Einav S, Quake SR | 2018 | Single-cell transcriptional dynamics of flavivirus infection | https://www.ncbi.nlm.nih.gov/geo/query/acc.cgi?acc=GSE110496 | NCBI Gene Expression Omnibus, GSE110496 |
| Tirosh I, Venteicher A, Suva M, Regev A | 2016 | Single-cell RNA-seq supports a developmental hierarchy in human oligodendroglioma | https://www.ncbi.nlm.nih.gov/geo/query/acc.cgi?acc=GSE70630 | NCBI Gene Expression Omnibus, GSE70630 |
| Amit I, Tanay A, Paul F, Arkin Y, Giladi A | 2015 | Transcriptional heterogeneity and lineage commitment in myeloid progenitors | https://www.ncbi.nlm.nih.gov/geo/query/acc.cgi?acc=GSE72857 | NCBI Gene Expression Omnibus, GSE72857 |
| Smyth GK, Chen Y, Pal B, Visvader JE | 2017 | Construction of developmental lineage relationships in the mouse mammary gland by single-cell RNA profiling | https://www.ncbi.nlm.nih.gov/geo/query/acc.cgi?acc=GSE95430 | NCBI Gene Expression Omnibus, GSE95430 |
| Häring M, Zeisel A, Linnarsson S, Ern-fors P | 2018 | Neuronal atlas of the dorsal horn defines its architecture and links sensory input to transcriptional cell types | https://www.ncbi.nlm.nih.gov/geo/query/acc.cgi?acc=GSE103840 | NCBI Gene Expression Omnibus, GSE103840 |
| Darmanis S, Enge | 2015 | A survey of human brain | https://www.ncbi.nlm. | NCBI Gene |

| | | | | | |
|---|---|---|---|---|---|
| M, Quake SR, Sloan SA, Barres BA, Zhang Y, Caneda C, Hayden Gephart MG, Shuer LM | | transcriptome diversity at the single cell level | nih.gov/geo/query/acc.cgi?acc=GSE67835 | Expression Omnibus, GSE67835 | |
| Wang YJ, Schug J, Golson ML, Won K, Liu C, Naji A, Avrahami D, Kaestner KH | 2016 | Single-cell transcriptomics of the human endocrine pancreas | https://www.ncbi.nlm.nih.gov/geo/query/acc.cgi?acc=GSE83139 | NCBI Gene Expression Omnibus, GSE83139 | |
| Veres A, Baron M | 2016 | A single-cell transcriptomic map of the human and mouse pancreas reveals inter- and intra-cell population structure | https://www.ncbi.nlm.nih.gov/geo/query/acc.cgi?acc=GSE84133 | NCBI Gene Expression Omnibus, GSE84133 | |
| Fincher CT, Wurtzel O, de Hoog T, Kravarik KM, Reddien PW | 2018 | Cell type transcriptome atlas for the planarian Schmidtea mediterranea | https://www.ncbi.nlm.nih.gov/geo/query/acc.cgi?acc=GSE111764 | NCBI Gene Expression Omnibus, GSE111764 | |
| Segerstolpe Å, Palasantza A, Eliasson P, Andersson EM, Andréasson AC, Sun X, Picelli S, Sabirsh A, Clausen M, BjursellMK, Smith DM, Kasper M, Ämmälä C, Sandberg R | 2016 | Single-cell transcriptome profiling of human pancreatic islets in health and type 2 diabetes | https://www.ebi.ac.uk/arrayexpress/experiments/E-MTAB-5061 | ArrayExpress, E-MTAB-5061 | |
| Muraro MJ, Dharmadhikari G, de Koning E, van Oudenaarden A | 2016 | A Single-cell transcriptome atlas of the human pancreas | https://www.ncbi.nlm.nih.gov/geo/query/acc.cgi?acc=GSE85241 | NCBI Gene Expression Omnibus, GSE85241 | |

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
