## [Decision Letter]

Thank you for submitting your article "Self-assembling manifolds in single-cell RNA sequencing data" for consideration by *eLife*. Your article has been reviewed by three peer reviewers, and the evaluation has been overseen by Alex K. Shalek as the Reviewing Editor and Naama Barkai as the Senior Editor. The following individual involved in the review of your submission has agreed to reveal their identity: Itai Yanai (Reviewer #1).

The reviewers have discussed the reviews with one another and the Reviewing Editor has drafted this decision to help you prepare a revised submission.

Summary:

Tarashansky et al. detail a computational algorithm called SAM (Self-Assembling Manifold) that helps to identify meaningful clusters within single-cell RNA-Seq data. SAM iteratively re-weights gene expression by importance (as determined by variation across neighborhoods rather than individual cells) until convergence; the resulting nearest neighbor graph is then used for clustering, and gene weights can be used to indicate biological significance. The authors benchmark SAM using previously published data as well as a self-generated Schistosoma dataset that had proven difficult to cluster using existing methods. Overall, the manuscript is well written and the algorithm has the potential to be impactful.

Essential revisions:

• The authors should provide better documentation on SAM to make the scripts more understandable to a general audience and aid in its application to new datasets. Specific guidelines on the selection of variable parameters (e.g., SAM weight cutoffs; illustrated through the presented examples) are particularly important, as is examination of the extent to which SAM is confounded by integrating data across batches and methods. More example applications should be included in the documentation, as well as in-depth function descriptions.

• A more comprehensive analysis of the parameter space for some of the existing methods (e.g., Seurat) should be performed to properly benchmark SAM. Utilizing data for which ground truth is known could be particularly illuminating. The authors should also discuss under what conditions SAM performs well to help guide its implementation.

---

## [Author Response]

Essential revisions:• The authors should provide better documentation on SAM to make the scripts more understandable to a general audience and aid in its application to new datasets. Specific guidelines on the selection of variable parameters (e.g., SAM weight cutoffs; illustrated through the presented examples) are particularly important, as is examination of the extent to which SAM is confounded by integrating data across batches and methods. More example applications should be included in the documentation, as well as in-depth function descriptions.

We have included numerous tutorials to the Github repository (https://github.com/atarashansky/self-assembling-manifold/tree/master/tutorial) describing in detail the various functions, parameters, attributes, data structures, and applications of the SAM package. Additionally, we have included documentation (docstrings) for all functions available to the users in the SAM package.

SAM is generally robust to the choice of parameters. All datasets in this study were analyzed using default parameters. Nevertheless, in the Materials and methods section we now explain that data preprocessing methods may affect SAM analysis. Briefly, we have found that standardization performs well with large, sparse datasets that are expected to contain many expected subpopulations, whereas L2-normalization is more suitable for smaller datasets with fewer subpopulations. We have also added an in-depth Jupyter notebook tutorial to the Github illustrating how the analysis might change when varying the parameters, which include data preprocessing method, number of nearest neighbors in the graph, distance metric, and gene weight normalization, although we have never needed to change these parameters in practice. Additionally, we have developed an interactive user interface that facilitates the convenient exploration of single cell data and SAM parameters (Figure 1—figure supplement 2). A Jupyter notebook tutorial explaining how to use the interface is provided as well.

Like almost all single-cell manifold reconstruction methods, SAM is susceptible to batch and cell cycle effects. However, we have shown examples (Figure 2f and Figure 6) that use existing batch correction method and gene set enrichment analysis in conjunction with SAM to correct for these effects. Additionally, we now provide detailed Jupyter notebooks recapitulating our analyses of the schistosome stem cell data, which contain two different batches, and activated macrophage data, which is severely biased by cell cycle effects, to the tutorial section of the Github repository.

• A more comprehensive analysis of the parameter space for some of the existing methods (e.g., Seurat) should be performed to properly benchmark SAM. Utilizing data for which ground truth is known could be particularly illuminating. The authors should also discuss under what conditions SAM performs well to help guide its implementation.

We ran a parameter sweep on the Seurat parameters including the number of Highly Variable Genes (HVGs) and the number of principal components to include in the analysis on each of the nine benchmarking datasets. For each dataset, we selected the parameters with the highest clustering accuracy measured by ARI and added these results to Figure 3a. Even with this extensive parameter sweeping, which is impossible to perform on an experimental dataset with no available ground truth labels, for five out of nine benchmarking datasets, SAM still has the highest clustering accuracy. On the remaining four datasets, both Seurat and SAM have near-perfect clustering accuracy. Furthermore, in the corruption test, Seurat run with optimized parameters is no more robust than Seurat run with default parameters (Figure 5).

With extensive benchmarking (Figure 3-5) on nine gold standard benchmarking datasets, another 47 experimental datasets, and numerous simulated datasets, there is sufficient evidence to support that SAM performs generally well applied to both ‘simple’ and ‘challenging’ datasets.